# CONDITIONAL DENSITY RATIO SCORE FOR POST HOC DEEP OUTLIER DETECTION

## ABSTRACT

The ability to accurately identify out-of-distribution (OOD) samples is essential not only as a stand-alone machine learning task but also for maintaining the reliability and safety of machine learning systems. Within this domain, post hoc density estimators like the energy score are popular ways for detecting OOD samples. However, most of the existing post hoc density estimation have mainly focused on marginalizing the conditional distributions over all possible classes. In this paper, we introduce the Conditional Density Ratio (CDR) score, a principled post hoc density estimator that leverages both a class-conditional generative model in the latent space and a discriminative classifier model, allowing us to estimate the marginal densities of the latent representation without marginalization. We demonstrate that a key component to the success of the CDR score lies in correctly calibrating the two models and propose a simple yet effective method to automatically tune the temperature parameter without the need for out-of-distribution samples. We illustrate the general compatibility of the proposed method with two popular density estimators, the kernel density estimator and the Mahalanobis estimator. Through experiments on a wide range of OOD benchmark tasks, we verify the effectiveness of the proposed method and advocate it as an easy-to-implement baseline that can achieve competitive performance in most tested scenarios.

## 1 INTRODUCTION

As the field of machine learning rapidly advances, ensuring the integrity and safety of predictive models becomes crucial, particularly when deployed in dynamic and uncertain environments. A fundamental aspect of ensuring model reliability is the effective detection of outlier/out-of-distribution (OOD) samples, which is pivotal not only for maintaining the robustness of ML systems but also for their capability to perform standalone outlier detection tasks. Traditional OOD detection techniques often fall short in complex and high-dimensional data scenarios. Following recent advancements in neural networks, a plethora of neural network-based OOD detection methods have been proposed (Yang et al., 2022; Han et al., 2022). Among these methods, post-hoc detection methods like the maximum softmax probability (MSP) score (Liu et al., 2020; Hendrycks & Gimpel, 2016; Liang et al., 2017; Huang et al., 2021; Lee et al., 2018) stand out as favorable in practice due to their plug-and-play simplicity with pre-trained neural networks. This is especially beneficial in the era of large-scale models where fine-tuning can be challenging without access to latest compute resources.

Recently, post-hoc density estimation based methods like the energy score (Liu et al., 2020) and the Gaussian mixture based energy measurement (GEM) score (Morteza & Li, 2022) have also been proposed for OOD detection. However, most of the existing post hoc density estimation have mainly focused on marginalizing the conditional distributions over all possible classes. In this paper, we explore an alternative approach, and propose a simple yet principled method termed the Conditional Density Ratio (CDR) score for post hoc density estimation. At a high level, the proposed CDR score leverages both a class-conditional generative model in the latent space and a discriminative classifier model to estimate the densities for outlier detection without the need for marginalization with respect to different classes. Importantly, the CDR score is a general framework that applied to a wide range of density estimators. In this paper, we showcase the effectiveness of the proposed method with the Mahalanobis estimator and the kernel density estimation. Like all post-hoc detection methods, the CDR score builds upon trained models without requiring additional training, enabling the utilization of both models with minimal computational overhead. The additional conditional generative models

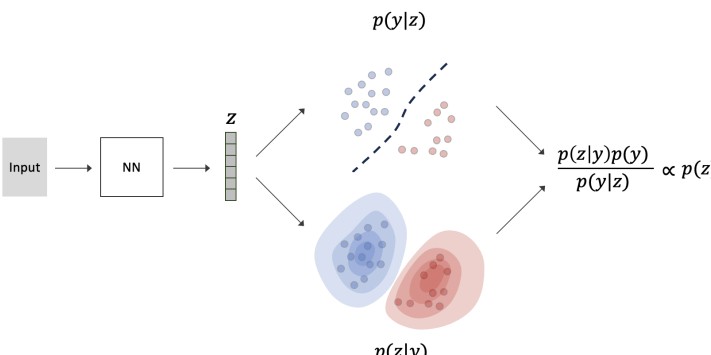

Figure 1: An illustration on the proposed Conditional Density Ratio score. Given an input, a pre-trained neural network is used to obtain the latent representation $\boldsymbol{z}$. Then, we approximate $p_\psi(\boldsymbol{z}|y)$ using models like a mixture of Gaussian, and use the pretrained neural network classifier for $p_\phi(y|\boldsymbol{z})$. Assuming a uniform prior $p(y)$, the conditional density ratio $\log \frac{p_\psi(\boldsymbol{z}|y)}{p_\phi(y|\boldsymbol{z})}$ is used for outlier detection.

in the latent space can be obtained using a small number of samples with minimal cost. A cornerstone of our proposed solution is a novel temperature scaling technique that effectively balances the relative contributions of the generative and discriminative models, optimizing their performance without needing access to any OOD samples for temperature tuning.

**Problem Setup**   To simulate a challenging real-world scenario, we consider a setup where we have access to the pretrained model weights but not the training data. Additionally, we have no access to any OOD samples and only a small number of in-distribution (ID) data for validation. Such a scenario is frequently encountered in real-world applications, yet it is under-explored in the context of outlier detection. We evaluate the effectiveness of the CDR score on a wide range of OOD detection benchmarks, demonstrating superior performance. With the temperature scaling technique, our approach is hyperparameter-free and easily adoptable for any OOD model, serving as a strong baseline in the domain of outlier detection.

**Summary of Contributions**   Firstly, we advocate for a previously under-emphasized post-hoc OOD detection setup where only the model weights and a small number of inlier validation samples are available. Secondly, we propose the Conditional Density Ratio (CDR) score, a general framework for post hoc OOD detection. Thirdly, we emphasize temperature scaling as a key component of the proposed method and devise a novel methodology to automatically tune the temperature parameters with a small number of inlier samples. Lastly, we showcase the wide applicability of the proposed method and empirically demonstrate its effectiveness.

## 2   RELATED WORKS

**Post-Hoc Methods**   Post-hoc OOD detection is a widely adopted setup for detecting out-of-distribution samples. A popular approach is the maximum softmax probability (MSP) score (Hendrycks & Gimpel, 2016). Building upon this approach, temperature scaling and adversarial perturbation have been found useful for MSP score-based OOD detection (Liang et al., 2017). Adversarial perturbation can be seen as implicitly leveraging gradient information for OOD detection, while another line of research focuses on explicitly using gradients as the score for detection (Huang et al., 2021; Behpour et al., 2024; Igoe et al., 2022). It has also been found that using cosine distance on the latent representation yields better OOD detection (Noh et al., 2023; Techapanurak et al., 2019). More recently, Liu & Qin (2023) has proposed a method to construct decision boundaries for OOD detection given a trained classifier. Alternatively, Shannon entropy can also be used as the score for post-hoc OOD detection (Liu et al., 2023). Most of the aforementioned methods use the trained classifier model for OOD detection. Instead, applying Mahalanobis distance measure in the latent representation space has also achieved success for OOD detection (Lee et al., 2018; Chen et al., 2023). However, none of these methods model the underlying $p(x)$ distribution for OOD, which can be considered a fundamental flaw, under reasonable assumptions. In this work,

we propose a unified approach that leverages both the discriminative classifier and the Mahalanobis distance in a principled way to approximate the underlying $p(x)$ for OOD detection. Most similar to our work are (Morteza & Li, 2022; Peng et al., 2024; Liu et al., 2020), which also highlight the limitations of MSP and Mahalanobis distance due to their underlying assumptions. However, instead of using conditional probability ratios to mitigate the issue, these methods propose to aggregate over all possible classes to approximate $p(x)$ instead. We empirically show that the performance of these methods falls short. Recently, the non-parametric KNN algorithm has also been demonstrated to work well for OOD detection (Sun et al., 2022). Lastly, we emphasize that, unlike many previously proposed methods which contain hyperparameters that are hard to tune without access to OOD data, our proposed method only includes a temperature parameter that we demonstrate can be effectively tuned with only a small number of inlier samples.

**Training-Time Adjustments** Aside from post-hoc OOD methods, many train-time adjustments have been proposed to tackle the problem of OOD detection. For instance, it was recently discovered that the norm of logits can be harmful for OOD detection, and a simple fix was proposed to normalize logits during the training of neural networks (Wei et al., 2022). Similarly, research has proposed normalizing the latent representation space before logits (Haas et al., 2023). Most of these methods require customized training of models, which can be a significant limitation when models are large and training data is not available. Furthermore, it has been demonstrated that outlier exposure during training can significantly enhance the performance of OOD detection (Hendrycks et al., 2018). In light of this observation, numerous methods have been proposed to exploit different types of outlier samples during training (Zheng et al., 2023; Zhu et al., 2023; Du et al., 2021; 2023; Tao et al., 2023). However, incorporating OOD samples during training can lead to bias in the model. In this work, we consider a setup where no labeled OOD samples are available for training nor inference.

**Unsupervised/Self-Supervised Methods** OOD detection has also been considered in the context of unsupervised learning where no label information is available. This line of research typically overlaps with the literature on anomaly detection (Salehi et al., 2021). Traditionally, such problems are tackled using support vector machine models (Fumera & Roli, 2002; Li et al., 2003). These models have recently been combined with neural networks (Ruff et al., 2018). Aside from discriminative approaches, generative unsupervised methods like normalizing flows (Chali et al., 2023), VAE (Xiao et al., 2020; An & Cho, 2015), and GANs (Di Mattia et al., 2019) have also been proposed for OOD detection. However, these generative models have generally shown poor OOD detection performance (Kirichenko et al., 2020; Zhang et al., 2021; Nalisnick et al., 2019). Instead of modeling the density function explicitly, another interesting line of work considers self-supervised learning for OOD detection. For instance, contrastive learning has been demonstrated to be an effective approach to enhance OOD detection performance (Ming et al., 2022; Sehwag et al., 2021). Transformation-based self-supervised methods have also shown improved performance compared to explicitly modeling the density (Shenkar & Wolf, 2021; Qiu et al., 2021; Golan & El-Yaniv, 2018; Bergman & Hoshen, 2020). In this work, we demonstrate that post-hoc density estimation methods can be integrated into these pseudo-label-based self-supervised learning methods at test time.

## 3 METHOD

### 3.1 PRELIMINARIES

**Problem Statement** Let $\boldsymbol{x} \in \mathbb{R}^d$ denote an input feature and $y \in \{1, \dots, K\}$ its corresponding label. Jointly, an input-label tuple $(\boldsymbol{x}, y)$ is a random sample from some joint distribution $P_{\boldsymbol{X},Y}$. We denote the marginal distributions of the random variables $\boldsymbol{X}$ and $Y$ by $P_{\boldsymbol{X}}$ and $P_Y$, respectively. In this work, we assume that a dataset $\mathcal{D}_{\text{train}}^N = \{(\boldsymbol{x}_i, y_i)\}_{i=1}^N$ is sampled i.i.d. from $P_{\boldsymbol{X},Y}$. A classifier $f_\theta : \mathbb{R}^d \to \mathbb{R}^K$ parameterized by $\theta$ is trained on $\mathcal{D}_{\text{train}}^N$ to approximate the underlying conditional posterior probability $p(y|\boldsymbol{x})$. We denote the approximate conditional distribution as $\hat{p}_\theta(y|\boldsymbol{x})$.

After training, we consider the challenging scenario where the training data $\mathcal{D}_{\text{train}}^N$ is no longer available. Instead, a validation set $\mathcal{D}_{\text{val}}^M$ consisting of samples drawn i.i.d. from $P_{\boldsymbol{X},Y}$ can be attained, where $M \ll N$. At test time, in addition to the inlier marginal distribution $P_{\boldsymbol{X}}$, samples can also be drawn from an outlier distribution $Q_{\boldsymbol{X}}$. The goal of post-hoc OOD detection is to leverage the learned (frozen) classifier $f_\theta$ to devise a binary classifier $g(\boldsymbol{x})$ to determine whether a test sample

$\boldsymbol{x}$ is sampled from $P_{\boldsymbol{X}}$ (ID) or $Q_{\boldsymbol{X}}$ (OOD). Specifically, given a threshold $\lambda$ and a scoring function $S(\boldsymbol{x})$, the decision can be made via a level set estimation:

$$g_\lambda(\boldsymbol{x}) = \begin{cases} \text{ID} & S(\boldsymbol{x}) \geq \lambda \\ \text{OOD} & S(\boldsymbol{x}) < \lambda. \end{cases}$$

Under the reasonable assumption that outlier samples have zero or low likelihood under the inlier distribution $P_{\boldsymbol{X}}$, a natural choice to the scoring function $S(\boldsymbol{x})$ is the ID density function $p(\boldsymbol{x})$(Zhang et al., 2021; Morteza & Li, 2022; Peng et al., 2024). We formalize the intuition in Appendix A.

We parameterize the classifier model $f_\theta$ with a neural network consisting of a multi-layer feature extractor $h_\theta(\boldsymbol{x})$ which maps input feature $\boldsymbol{x}$ to a low dimensional feature $\boldsymbol{z}$, followed by a linear classifier $f_\psi$ that maps $\boldsymbol{z}$ to the logit output. Since $f_\theta$ is a fixed deterministic function at test time, we make the assumption that $\boldsymbol{z}$ is a faithful representation of $\boldsymbol{x}$, and will use $\boldsymbol{z}$ interchangeably with $\boldsymbol{x}$.

**Maximum Softmax Probability for OOD**  Given a feature extractor $h_\theta$ that maps $\boldsymbol{x}$ to a low dimensional feature $\boldsymbol{z}$, and a linear classifier $f_\psi$ which maps $\boldsymbol{z}$ to logits, a straightforward approach is the maximum softmax probability (MSP) (Hendrycks & Gimpel, 2016). Formally,

$$MSP(\boldsymbol{z}; \psi, T) = \max_i \hat{p}_\psi(y|\boldsymbol{z}) = \max_i \frac{\exp\left(f_{\psi_i}(\boldsymbol{z})/T\right)}{\sum_j \exp\left(f_{\psi_j}(\boldsymbol{z})/T\right)} \approx \max_i p(y|\boldsymbol{z}) \not\propto p(\boldsymbol{z}),$$

where $T$ denotes the temperature constant that adjusts the spikiness of the output distribution (Guo et al., 2017; Liang et al., 2017). $T = 1$ is used in the original work (Hendrycks & Gimpel, 2016). A follow-up work proposes tuning of the temperature with a held-out OOD dataset (Liang et al., 2017). Such a method would not work for our problem setup. In addition, note that the MSP score does not equal the marginal density $p(\boldsymbol{z})$, potentially leading to suboptimal performance. Indeed, while a well-calibrated probability $p(y|\boldsymbol{x})$ captures the uncertainty inherent in the labeling space given an input data $\boldsymbol{x}$ drawn from the ID distribution, the underlying conditional $p(y|\boldsymbol{x})$ may be undefined when the sample $\boldsymbol{x}$ is drawn from the OOD distribution, where $p(\boldsymbol{x}) = 0$.

**Mahalanobis Distance for OOD**  Recently, Lee et al. (Lee et al., 2018) proposed to use Mahalanobis distance as the score function instead. Specifically, given the latent representation $\boldsymbol{z}$, a class-conditional distribution $p(\boldsymbol{z}|y)$ is approximated with a Gaussian distribution $\hat{p}_\phi(\boldsymbol{z}|y = c) = \mathcal{N}(\boldsymbol{z}|\mu_c, \boldsymbol{\Sigma})$, where $\phi = \{\mu_1, \ldots, \mu_k, \boldsymbol{\Sigma}\}$. Under this assumption, the Mahalanobis distance (Hastie & Tibshirani, 1996) score function is

$$M(\boldsymbol{z}; \phi) = \max_i \exp\left(-\frac{1}{2}(\boldsymbol{z} - \mu_i)^\top \boldsymbol{\Sigma}^{-1}(\boldsymbol{z} - \mu_i)\right)$$
$$\propto \max_i \hat{p}_\phi(\boldsymbol{z}|y = i) \approx \max_i p(\boldsymbol{z}|y = i) \not\propto p(\boldsymbol{z}).$$

Under the scenario where $p(y|\boldsymbol{z})$ is degenerate at the ground truth label, the Mahalanobis distance becomes equivalent to the marginal density $p(\boldsymbol{z})$. However, this scenario may not hold even for ID samples, which means the Mahalanobis distance can be sub-optimal in practice.

**Post Hoc Density Estimation for OOD**  Given a pretrained classifier network, it is possible to approximate the marginal density $p(\boldsymbol{z})$. One example is the energy score (Liu et al., 2020):

$$E(\boldsymbol{z} : \psi, T) = \log \sum_i -\exp\left(f_{\psi_i}(\boldsymbol{z})/T\right) \propto \log \sum_i \hat{p}_\psi(\boldsymbol{z}|y = i) = \log \hat{p}_\psi(\boldsymbol{z}) \approx \log p(\boldsymbol{z}),$$

under the assumption that the partition function $Z(\psi)$ of $\hat{p}_\psi(\boldsymbol{z}|y) = \frac{\exp\left(f_\psi(\boldsymbol{z})/T\right)}{Z(\psi)}$ for normalization is a constant (Grathwohl et al., 2019). This may not hold in practice.

Alternatively, for Mahalanobis distance, density estimation can be achieved with the GEM (Gaussian mixture based Energy Measurement) score (Morteza & Li, 2022)

$$GEM(\boldsymbol{z}; \phi, T) = \log \sum_i \exp\left(-\frac{1}{2T}(\boldsymbol{z} - \mu_i)^\top \boldsymbol{\Sigma}^{-1}(\boldsymbol{z} - \mu_i)\right) \propto \log \sum_i \hat{p}_\phi(\boldsymbol{z}|y_i) \approx \log p(\boldsymbol{z}),$$

under uniform class prior $p(y)$. The partition function for normalization is a constant in this case.

**Remark**  It is unconventional to introduce a temperature term into Mahalanobis distance and Gaussian distribution. Previously, $T = 1$ has been used for both GEM and Mahalanobis distance for OOD. However, as we demonstrate below, the temperature term is a crucial for our proposed method.

## 3.2 Conditional Density Ratio for Anomaly Detection

Prior works on post hoc density estimation have mainly focused on summing of conditional distributions over all possible classes. In this work, we consider an alternative method to obtain an density estimation given a pre-trained classifier network. Specifically, by law of conditional probability, we have $p(\boldsymbol{z}) = \frac{p(\boldsymbol{z}|y)p(y)}{p(y|\boldsymbol{z})}$. Given a class-conditional approximations $\hat{p}_\phi(\boldsymbol{z}|y)$, we can obtain $\hat{p}_\phi(y|\boldsymbol{z})$ by Bayes rule, resulting in a trivial solution of $p(\boldsymbol{z}) \approx \hat{p}_\phi(\boldsymbol{z})$. However, if we instead approximate $p(y|\boldsymbol{z})$ with the classifier head $\hat{p}_\psi(y|\boldsymbol{z})$, the marginal density can be approximated by $\frac{\hat{p}_\phi(\boldsymbol{z}|y)p(y)}{\hat{p}_\psi(y|\boldsymbol{z})}$. Assuming a uniform class prior $p(y)$, we have the Conditional Density Ratio (CDR) score

$$CDR_i(\boldsymbol{z}; \phi, \psi) = \log \hat{p}_\phi(\boldsymbol{z}|y=i) - \log \hat{p}_\psi(y=i|\boldsymbol{z})$$

$$\propto \left( -\frac{1}{2T_\phi}(\boldsymbol{z} - \mu_i)^\top \boldsymbol{\Sigma}^{-1}(\boldsymbol{z} - \mu_i) \right) - \log \frac{\exp\left(f_{\psi_i}(\boldsymbol{z})/T_\psi\right)}{\sum_j \exp\left(f_{\psi_j}(\boldsymbol{z})/T_\psi\right)}$$

$$\propto \log \hat{p}_{\phi,\psi}(\boldsymbol{z}) \approx \log p(\boldsymbol{z}).$$

The proposed CDR score is a general framework that can be applied with a wide range of density estimators. Building on the prior success of GEM for OOD detection, a natural choice is to estimate $p(\boldsymbol{z}|y)$ using a mixture of Gaussian distributions, denoted as $\hat{p}_\phi(\boldsymbol{z}|y)$. Consequently, the resulting CDR score provides a principled way to combine the key components of GEM and energy scores.

Unlike the energy and GEM scores, which derive marginal densities by marginalizing over $y$, the CDR score estimates $p(\boldsymbol{z})$ by conditioning on $y$. Therefore, for a $K$-class classification problem, there are $K$ distinct CDR scores. With perfect estimation of the conditional probabilities, all of these CDR scores will converge to the same value.In practice, CDR scores obtained conditioned on different classes can vary. We adopt a simple solution to aggregate the CDR scores across different class, giving us

$$CDR(\boldsymbol{z}; \phi, \psi) = \frac{1}{K} \sum_i^k CDR_i(\boldsymbol{z}; \phi, \psi). \tag{1}$$

**On Marginalization Versus Conditional Density Ratio**   While the CDR score does not offer a computational advantage over marginalization-based methods such as the energy score or GEM score, we empirically demonstrate in the experimental section below that the proposed CDR score performs competitively against its marginalization-based counterparts. We attribute this to the fact that the CDR score leverages a combination of two distributions. Specifically, the CDR score can be interpreted as a principled approach to combining $p(y|\boldsymbol{z})$ and $p(\boldsymbol{z}|y)$, both of which have been independently used for OOD detection, to achieve better performance when used together.

From a theoretical perspective, the energy score is only a valid density estimator if the partition function for normalization remains constant. However, this assumption may not hold in practice, potentially leading to inaccurate estimates. In sharp contrast, while the same classifier $\hat{p}_\psi(y|\boldsymbol{z})$ is used, the CDR score avoids marginalization, allowing it to serve as a valid density estimator even in scenarios where the partition function for normalization is not constant.

**The relationship between CDR and Energy Score**   To gain further insight into what the CDR score is doing, we can rewrite the CDR score as

$$CDR(\boldsymbol{z}; \phi, \psi) \propto \underbrace{\frac{1}{K} \sum_i^k \log \hat{p}_\phi(\boldsymbol{z}|y=i) - \log \hat{p}_\psi(\boldsymbol{z}|y=i)}_{\text{Average Generative Conditional Ratio}} + \underbrace{\log \sum_j \hat{p}_\psi(\boldsymbol{z}|y=j)}_{\text{Energy Score}}, \tag{2}$$

As such, the CDR score can be interpreted as a correction to the energy score. As we demonstrate in Section 5, with correctly scaled temperatures, the average generative conditional ratio (GCR) is much closer to zero for inlier samples. Indeed, under perfect estimation, the GCR is trivially zero for all classes. Thus, combining GCR with energy score yields superior OOD performance.

**Setting Temperatures for CDR Score**   A crucial component of the success of the CDR score is setting the temperatures $T_\phi$ and $T_\psi$ effectively. One can easily scale the temperatures by optimizing the separation between ID and OOD data with access to OOD data (Liang et al., 2017; Hendrycks

---

**Algorithm 1** Temperature Scaling for CDR Score

---

**Input** Small Validation dataset $D_{\text{val}} = \{(\boldsymbol{z}_i, y_i)\}_{i=1}^N$, $\phi$ and $\psi$.

    Initialize $T_\psi = 1$, $T_\phi = 1$.

    Update $T_\psi = \arg\min_{T_\psi} - \sum_i \log \hat{p}_\psi(y_i|\boldsymbol{z}_i)$ [Temperature Scaling for $\hat{p}_\psi(y|\boldsymbol{z})$]

    Update $T_\phi = \arg\min_{T_\phi} \sum_i - \log\left(\hat{p}_\phi(\boldsymbol{z}_i|y = y_i) - \frac{1}{K-1}\sum_{j \neq y_i}\hat{p}_\phi(\boldsymbol{z}_i|y = j)\right) + \mathcal{R}(\boldsymbol{z}_i, y_i|T_\psi)$

    [Temperature Scaling for $\hat{p}_\phi(\boldsymbol{z}|y)$]

**Output** $T_\psi$, $T_\phi$

---

et al., 2018). However, it becomes much more challenging to find the right temperatures with only ID data. Our guiding principle in selecting the right temperatures lies in maximizing the likelihood of the density functions on the inlier data. For $\hat{p}_\psi(y|\boldsymbol{z})$, this can be achieved by minimizing the negative log-likelihood function with respect to $T_\psi$. This approach, commonly used to calibrate neural networks for classification (Guo et al., 2017), can also benefit outlier detection. Intuitively, a more calibrated $\hat{p}_\psi(y|\boldsymbol{z})$ yields a better density estimator and, hence, improved OOD detection performance. Inspired by this, we propose a novel strategy for selecting $T_\phi$. Given a labeled sample $(\boldsymbol{z}_i, y_i)$, we maximize the likelihood $\hat{p}_\phi(\boldsymbol{z}_i|y = y_i)$ for the ground-truth label while minimizing the likelihood $\hat{p}_\phi(\boldsymbol{z}_i|y = j)$ for all $j \neq y_i$. Naively optimizing the two temperatures independently can result in density functions that are orders of magnitude different from each other, leading to suboptimal performance. To unleash the full power of the log ratio, we add a regularization term when optimizing $T_\phi$ so that the two components $\hat{p}_\phi(\boldsymbol{z}|y)$ and $\hat{p}_\psi(y|\boldsymbol{z})$ are comparable in scale on average. Formally, given a validation dataset $\mathcal{D}_{\text{val}} = \{(\boldsymbol{z}_i, y_i)\}_{i=1}^N$ and a temperature-tuned $\hat{p}_\psi(y|\boldsymbol{z})$, we optimize $T_\phi$ with the following approach:

$$\mathcal{L}\left(T_\phi|\mathcal{D}_{val}, T_\psi\right) = \sum_i - \log\left(\hat{p}_\phi(\boldsymbol{z}_i|y = y_i) - \frac{1}{K-1}\sum_{j \neq y_i}\hat{p}_\phi(\boldsymbol{z}_i|y = j)\right) + \mathcal{R}(\boldsymbol{z}_i, y_i|T_\psi), \quad (3)$$

where $\mathcal{R}(\boldsymbol{z}_i, y_i) = \frac{1}{K}\sum_{j=1}^K |\log \hat{p}_\phi(\boldsymbol{z}_i|y = j) - \log \hat{p}_\psi(y = j|\boldsymbol{z}_i)|$. In practice, the above optimization can be solved efficiently using grid search, since the logits of the samples can be pre-computed and saved. We illustrate the procedure in Algorithm 1.

**Extension to Kernel Density Estimators (KDEs)** The CDR score is generally applicable to a wide range of density estimators. In this paper, we also explore the Parzen-Rosenblatt window method (Hastie et al., 2009) as an alternative to approximate $p(\boldsymbol{z}|y)$. Under such an assumption,

$$\hat{p}(\boldsymbol{z}|y = i) \propto \sum_j \exp\left(-\frac{\left(\boldsymbol{Z}_j^i - \boldsymbol{z}\right)^T \boldsymbol{\Sigma}^{-1}\left(\boldsymbol{Z}_j^i - \boldsymbol{z}\right)}{T}\right), \quad (4)$$

where $\boldsymbol{Z}_1^i \ldots \boldsymbol{Z}_n^i$ corresponds to all the samples in the validation set with label $y = i$. We empirically observed that the Mahalanobis distance-based structured kernel yields better performance compared to the simple Gaussian kernel. For consistency, the window size is replaced with the temperature term $T$. We use the same strategy as above to search for $T$ automatically using validation set.

We note that the KDE estimator is more computationally expensive than the Mahalanobis estimator. Specifically, during inference, the computational complexity of the KDE estimator scales linearly with the number of samples. This can become computationally intensive when the sample size for KDE computation is large. Therefore, in all our experiments, we use a very small validation set to compute the KDE score. Further details are provided in the experimental section below. Additionally, since the KDE score is computed in the latent representation space $\boldsymbol{z}$, we precompute and cache the latent representation vectors $\boldsymbol{z}$ for all validation samples immediately after training. This approach makes the KDE estimator practical for real-world applications.

**Extension to Self-Supervised Learning** The CDR score can also be applied to self-supervised scenarios. For instance, rotation prediction given a dataset of unlabeled images has found to be a useful self-supervised learning task (Golan & El-Yaniv, 2018) for outlier detection. Instead of using the MSP score, we can similarly apply the CDR score. A significant advantage of the CDR score compared to energy score is the reduced reliance on the accuracy of the classifier network

for detection, provided the classifier is calibrated. Given a less confident classifier prediction, the CDR score will automatically rely more on the Mahalanobis distance for detection. This is beneficial especially for the case of self-supervised learning where even the underlying $p(y|\boldsymbol{z})$ can have high uncertainty. In the extreme scenario when $p(y|\boldsymbol{z}) = p(y)$, energy score-based OOD detection becomes a coin flip, and CDR reduces to the GEM score with a uni-modal Gaussian distribution.

## 4 EXPERIMENTS

**Benchmark Methods** We compare the CDR score with MSP score (Hendrycks & Gimpel, 2016), Energy score (Liu et al., 2020), Mahalanobis distance score (Lee et al., 2018), GEM score (Morteza & Li, 2022) and GradNorm score (Huang et al., 2021). We have selected these methods for benchmark comparison as all of these are post-hoc OOD detection methods that do not require additional training nor auxilliary data (Yang et al., 2022). To compare with the baseline methods in their original configurations, we set temperatures to 1 for all the baseline methods. In Appendix E, we evaluate the impact of temperatures on these benchmark methods. To further illustrate the effectiveness of the proposed method, we also benchmark against the KNN score (Sun et al., 2022), a competitive baseline for post-hoc OOD detection with a hyper-parameter $k$. We use the $k = 50$ for all our experiments, a good hyper-parameter that worked well for a range of tasks in the original paper, We compute KNN scores using a small validation set for fair comparison. The goal of the experiments is to demonstrate that the proposed method can serve as an off-the-shelf competitive method that works on a wide range of datasets. As such, we have excluded comparisons against other relevant, but more complicated methods that require additional model training, finetuning and/or hyperparameter tuning (Liang et al., 2017; Hsu et al., 2020; Peng et al., 2024). We compare different methods with the false positive rate of OOD samples when true positive rate of ID samples is at 95% (**FPR95**) and the area under the ROC curve (**AUROC**) (Hendrycks & Gimpel, 2016).

**Datasets** We use common OOD benchmark datasets to demonstrate the effectiveness of our proposed method (Yang et al., 2022; Sun et al., 2022). Specifically, we use the CIFAR (Krizhevsky et al., 2009) datasets and the ImageNet dataset (Deng et al., 2009) as the ID data. For CIFARs, we use Textures (Cimpoi et al., 2014), SVHN (Netzer et al., 2011), LSUN-resize (Yu et al., 2015) and iSUN (Xu et al., 2015) as the OOD datasets. To further validate the effectiveness of the proposed method, we also use the hard-OOD datasets comprising of ImageNet-Resize, ImageNet-Fix and LSUN-fix and CIFAR10/CIFAR100 depending on the ID data (Tack et al., 2020). For ImageNet, following prior literature (Sun et al., 2022), we use Places365 (Zhou et al., 2017), Textures (Cimpoi et al., 2014), iNaturalist (Van Horn et al., 2018) and SUN (Xiao et al., 2010) as OOD datasets.

**Model Details** For CIFARs, we train our own classifier model because we need an additional validation set for our experiments. No hyperparameter tuning is done in training the classifier model. Specifically, we use a 121 layer DenseNet (Huang et al., 2017) as the classifier model for training. During training, we withhold 10% of the training dataset as the validation set. The same validation set is used at test time to compute temperatures and the means and variance for the Mahalanobis distance. We train the models using SGD with momentum 0.9, and weight decay $5 \times 10^{-4}$ for 200 epochs[1]. For ImageNet, we use the pretrained WideResNet50 (Zagoruyko & Komodakis, 2016) model provided by PyTorch. We use 25 samples per class to estimate the means and the covariance matrix, and for temperature tuning.

**Additional Experiments** To test the applicability of the proposed method on other models, we have conducted additional experiments using the ResNet model for CIFARs in Appendix F. We also benchmark the OOD detection performance of the proposed method on a wide range of models trained with the ImageNet dataset in Appendix D. Lastly, experiments with NLP datasets can be found in Appendix C. In general, similar conclusions hold across all experiments conducted.

### 4.1 RESULTS

We report experimental results on the supervised model in Tables 1, 2 and 3. To compare with the methods proposed in the original papers, temperatures are only applied to the CDR scores.

---

[1] We trained the model with code obtained at https://github.com/kuangliu/pytorch-cifar.

Table 1: OOD detection results on labeled CIFAR10 dataset using Desenset. "Maha" indicates Mahalanobis distance, "CDR_KDE" and "CDR_Maha" refers to CDR scores computed using kernel density estimator and the Mahalanobis distance respectively. **Bold** results indicate best performances.

| OOD | Average | | SVHN | | LSUN-Resize | | Texture | | iSUN | |
|---|---|---|---|---|---|---|---|---|---|---|
| | FPR95↓ | AUROC↑ | FPR95↓ | AUROC↑ | FPR95↓ | AUROC↑ | FPR95↓ | AUROC↑ | FPR95↓ | AUROC↑ |
| GradNorm | 69.65 | 65.86 | 66.4 | 70.84 | 66.5 | 69.99 | 79.06 | 53.14 | 66.62 | 69.45 |
| MSP | 57.61 | 90.62 | 63.33 | 89.82 | 49.65 | 92.87 | 65.25 | 87.43 | 52.19 | 92.36 |
| Energy | 37.26 | 91.99 | 46.27 | 90.88 | 22.25 | 96.02 | 57.02 | 85.43 | 23.48 | 95.62 |
| Maha | 20.35 | 95.58 | 22.54 | 95.31 | 17.14 | 96.76 | 23.48 | 93.88 | 18.24 | 96.38 |
| GEM | 20.38 | 95.58 | 22.58 | 95.3 | 17.17 | 96.75 | 23.53 | 93.87 | 18.25 | 96.38 |
| KNN | 24.95 | 95.67 | 33.71 | 94.89 | 15.39 | 97.19 | 33.69 | 93.77 | 16.99 | 96.82 |
| CDR_KDE | 14.68 | 97.01 | 19.71 | 96.51 | 7.17 | 98.59 | 23.94 | 94.52 | 7.9 | 98.43 |
| CDR_Maha | **10.53** | **97.72** | **11.87** | **97.75** | **5.37** | **98.8** | **18.42** | **95.7** | **6.47** | **98.64** |
| Hard-OOD | Average | | ImageNet-Resize | | ImageNet-Fix | | LSUN-fix | | CIFAR100 | |
| | FPR95↓ | AUROC↑ | FPR95↓ | AUROC↑ | FPR95↓ | AUROC↑ | FPR95↓ | AUROC↑ | FPR95↓ | AUROC↑ |
| GradNorm | 82.60 | 54.50 | 70.62 | 65.79 | 85.69 | 50.78 | 87.03 | 50.71 | 87.05 | 50.7 |
| MSP | 63.89 | 88.00 | 57.78 | 90.77 | 65.02 | 87.53 | 64.98 | 88.07 | 67.76 | 85.61 |
| Energy | 47.86 | 88.88 | 34.52 | 93.22 | 50.4 | 88.18 | 49.73 | 89.01 | 56.77 | 85.12 |
| Maha | 61.57 | 81.82 | 20.2 | 95.69 | 72.37 | 78.79 | 75.79 | 79.43 | 77.91 | 73.38 |
| GEM | 61.58 | 81.81 | 20.19 | 95.69 | 72.37 | 78.78 | 75.82 | 79.41 | 77.93 | 73.37 |
| KNN | 43.95 | 90.64 | 25.34 | 95.19 | 45.44 | **90.48** | 49.59 | 89.24 | **55.41** | **87.64** |
| CDR_KDE | **38.65** | **91.05** | 13.33 | 97.26 | **43.85** | 90.13 | **44.44** | **90.62** | 52.97 | 86.19 |
| CDR_Maha | 42.36 | 90.21 | **10.23** | **97.91** | 49.82 | 89.15 | 50.72 | 89.53 | 58.67 | 84.24 |

Table 2: OOD detection results on labeled CIFAR100 dataset with a pretrained Desenset.

| OOD | Average | | SVHN | | LSUN-Resize | | Texture | | iSUN | |
|---|---|---|---|---|---|---|---|---|---|---|
| | FPR95↓ | AUROC↑ | FPR95↓ | AUROC↑ | FPR95↓ | AUROC↑ | FPR95↓ | AUROC↑ | FPR95↓ | AUROC↑ |
| GradNorm | 87.53 | 51.58 | 74.37 | 75.06 | 96.48 | 35.41 | 83.21 | 59.47 | 96.06 | 36.37 |
| MSP | 84.42 | 73.71 | 83.96 | 75.66 | 83.09 | 74.28 | 86.76 | 72.21 | 83.85 | 72.69 |
| Energy | 82.09 | 77.20 | 83.75 | 77.67 | 77.44 | 80.75 | 87.32 | 71.73 | 79.85 | 78.65 |
| Maha | 35.62 | 91.68 | 62.36 | 85.88 | 23.66 | 94.95 | 30.34 | 91.86 | 26.1 | 94.03 |
| GEM | 35.65 | 91.67 | 62.42 | 85.86 | 23.67 | 94.95 | 30.35 | 91.84 | 26.17 | 94.03 |
| KNN | 44.15 | 90.28 | 48.5 | 91.25 | 47.54 | 90.08 | 34.34 | 90.72 | 46.23 | 89.07 |
| CDR_KDE | 30.52 | 92.69 | 52.29 | 87.72 | 18.68 | 95.47 | 29.65 | 92.83 | 21.47 | 94.73 |
| CDR_Maha | **19.55** | **96.01** | **37.88** | **93.19** | **9.27** | **97.84** | **19.27** | **95.81** | **11.79** | **97.19** |
| Hard-OOD | Average | | ImageNet-Resize | | ImageNet-Fix | | LSUN-fix | | CIFAR10 | |
| | FPR95↓ | AUROC↑ | FPR95↓ | AUROC↑ | FPR95↓ | AUROC↑ | FPR95↓ | AUROC↑ | FPR95↓ | AUROC↑ |
| GradNorm | 92.15 | 49.30 | 95.06 | 35.49 | 88.11 | 58.62 | 95.35 | 45.73 | 90.09 | 57.35 |
| MSP | 85.73 | 72.39 | 85.49 | 70.91 | 83.75 | 74.05 | **88.97** | **70.04** | 84.71 | 74.55 |
| Energy | 84.05 | 74.06 | 82.23 | 75.72 | 79.6 | 76.68 | 91.04 | 68.97 | **83.34** | **74.85** |
| Maha | 78.47 | 62.22 | 25.23 | 93.84 | 92.69 | 56.22 | 97.07 | 53.76 | 98.87 | 45.07 |
| GEM | 78.47 | 62.19 | 25.23 | 93.83 | 92.7 | 56.18 | 97.07 | 53.73 | 98.89 | 45.02 |
| KNN | 83.18 | 56.56 | 42.47 | 90.75 | 93.04 | 54.36 | 99.19 | 34.71 | 98.01 | 46.4 |
| CDR_KDE | **69.87** | **77.91** | 23.85 | 94.06 | **77.22** | **77.29** | 90.77 | 67.98 | 87.65 | 72.3 |
| CDR_Maha | 71.33 | 75.90 | **12.34** | **97.26** | 84.51 | 74.15 | 93.06 | 65.66 | 95.42 | 66.52 |

Table 3: OOD detection results on ImageNet dataset with a pretrained Wide-ResNet50.

| | Average | | Places | | SUN | | iNaturalist | | Texture | |
|---|---|---|---|---|---|---|---|---|---|---|
| | FPR95↓ | AUROC↑ | FPR95↓ | AUROC↑ | FPR95↓ | AUROC↑ | FPR95↓ | AUROC↑ | FPR95↓ | AUROC↑ |
| MSP | 61.07 | 84.29 | 67.75 | 82.41 | 64.97 | 83.5 | 46.58 | 90.04 | 64.98 | 81.21 |
| Energy | 60.55 | 86.31 | 66.12 | 84.32 | 61.26 | 86.21 | 55.73 | 90.49 | 59.08 | 84.20 |
| Maha | 79.52 | 74.83 | 94.4 | 65.87 | 92.97 | 67.33 | 87.74 | 75.42 | 42.98 | 90.68 |
| GEM | 79.52 | 74.83 | 94.4 | 65.87 | 92.97 | 67.33 | 87.74 | 75.42 | 42.98 | 90.68 |
| KNN | 52.22 | 87.24 | 76.42 | 78.9 | 71.39 | 82.26 | 47.49 | 91.32 | **13.58** | **96.47** |
| CDR_KDE | **45.61** | **91.02** | **63.22** | **86.07** | **59.36** | **88.39** | **43.74** | **93.47** | 16.13 | 96.15 |
| CDR_Maha | 64.82 | 85.61 | 83.66 | 78.62 | 81.9 | 81.08 | 71.16 | 86.74 | 22.55 | 96.00 |

**CDR achieves competitive performance** The proposed CDR score is the best-performing method on average among all the hyperparameter-free methods tested. Unlike MSP and Energy scores, which rely more heavily on the accuracy of the classifier model, CDR is much more robust even when the classifier does not perform as well. This is evident by comparing the performances of OOD detection when the ID distribution is CIFAR-10 versus CIFAR-100, the latter of which has a lower classifier performance. For example, the average AUROC for the Energy score dropped from 91.99% to 77.20% on the same set of OOD datasets. In contrast, the average AUROC for CDR_Maha only slightly decreased from 97.41% to 95.66%. Moreover, both CDR_KDE and CDR_Maha achieve competitive performances overall, indicating that the CDR framework is a general method for outlier detection. Comparing the two, CDR_KDE appears to be a more robust method, achieving

Table 4: Average outlier detection results on the self-supervised rotation prediction task.

| | CIFAR10 | | | | CIFAR100 | | | |
| | Average OOD | | Average Hard OOD | | Average OOD | | Average Hard OOD | |
| | FPR95↓ | AUROC↑ | FPR95↓ | AUROC↑ | FPR95↓ | AUROC↑ | FPR95↓ | AUROC↑ |
|---|---|---|---|---|---|---|---|---|
| GradNorm | 44.60 | 90.69 | 64.76 | 81.82 | 91.30 | 65.41 | 95.85 | 51.73 |
| MSP | 50.87 | 91.07 | 68.65 | 82.82 | 87.85 | 71.54 | 94.07 | 55.85 |
| Energy | 41.10 | 92.63 | 62.56 | 84.35 | 87.51 | 72.30 | 94.28 | 56.27 |
| Maha | 46.84 | 87.45 | 79.77 | 61.22 | 53.87 | 80.98 | 78.85 | 61.72 |
| GEM | 46.94 | 87.39 | 79.82 | 61.13 | 54.01 | 80.78 | 78.91 | 61.68 |
| KNN | 59.03 | 88.32 | 74.29 | 78.93 | 70.62 | 80.43 | 84.78 | **64.69** |
| CDR_KDE | 32.55 | 94.19 | 58.83 | **85.12** | 53.72 | 83.47 | 81.01 | 60.86 |
| CDR_Maha | **16.59** | **96.83** | **58.09** | 84.84 | **37.89** | **90.35** | **75.89** | 63.47 |

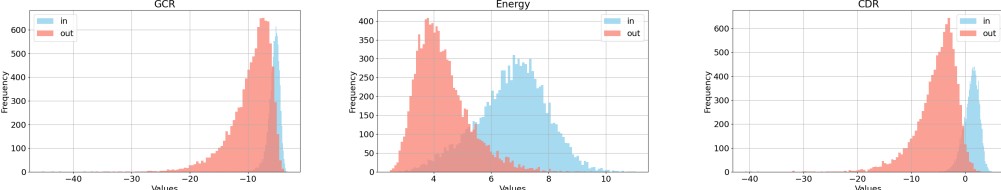

Figure 2: Histograms of the average generative conditional ratio (GCR) score, the energy score and the CDR score of a DenseNet model trained on CIFAR10 dataset. CDR score is a sum of the GCR and energy score. See Equation 2 for details. "In" and "Out" corresponds to ID and OOD samples respectively. Imagenet-Resize is used as the OOD dataset.

consistently competitive detection performance across a wide range of models and tasks. However, using CDR_KDE comes at the cost of increased computational requirements.

**When Does CDR Fail?**   We highlight a scenario where the CDR score performs subpar. As can be seen from the results on the hard-OOD dataset for CIFAR-100, when Mahalanobis distance performs poorly, both CDR methods are worse off than the Energy score. Comparatively, CDR_KDE is more robust. Note that, despite a chance-level detection performance by the Mahalanobis distance score, the CDR_Maha still recovers and achieves a reasonable OOD detection rate without the need for any hyperparameter tuning. Therefore, we promote the CDR and density ensemble as off-the-shelf metrics that can achieve competitive performance across a wide range of tasks.

**Density Scores Improves Self-Supervised Learning**   Lastly, we show in Table 4 the CDR score can also be applied to outlier detection with rotation prediction as the self-supervised learning task for improved detection performance. More details can be found in Appendix G.

## 5 ABLATION STUDY

**Breaking Down CDR Score** We have broken down in Equation 2 the CDR score into the average generative conditional ratio (GCR) and the energy score. To further gain empirical insights into the breakdown, we illustrate, using the CIFAR10 as the inlier distribution and the Imagenet-Resize dataset as the outlier distribution, the OOD detection performance of GCR, energy, and the CDR score. We plot the histogram of the scores in Figure 2. We highlight several observations. Firstly, the GCR score for the inlier dataset is much closer to zero for inlier samples on average. Indeed, under perfectly learned density functions, GCR scores are zero for all samples. In practice, due to estimation errors, GCR will be non-zero. Nevertheless, intuitively, the difference will be much smaller for inlier samples. Secondly, though the OOD detection performance of GCR alone is unsatisfactory, it can be combined with the energy score for much better performance.

**A Closer Look at Temperature Scaling for CDR Score**   Firstly, we compare AUROCs with and without the proposed temperature scaling. Results are summarized in Figure 3. As seen clearly, the proposed method for temperature scaling works well for most of the cases, often increasing the OOD detection performance significantly. Even in cases where temperature-scaled models perform subpar, the differences are small. To further understand how far away the temperature tuned with the proposed method is from the optimal temperature, we show plots of AUROC vs. temperature

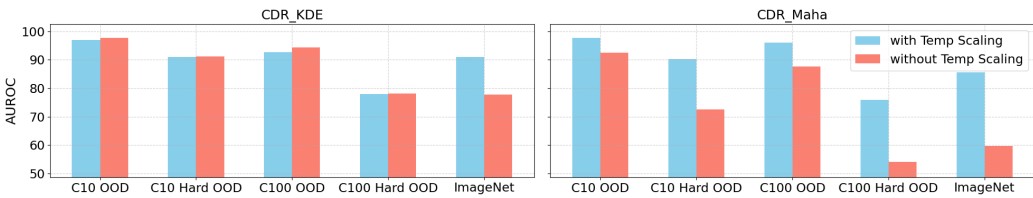

Figure 3: Bar plots of AUROCs of OOD detection for different methods and datasets with/without temperature scaling. Results obtained with DenseNet for CIFARs and Wide-ResNet for ImageNet.

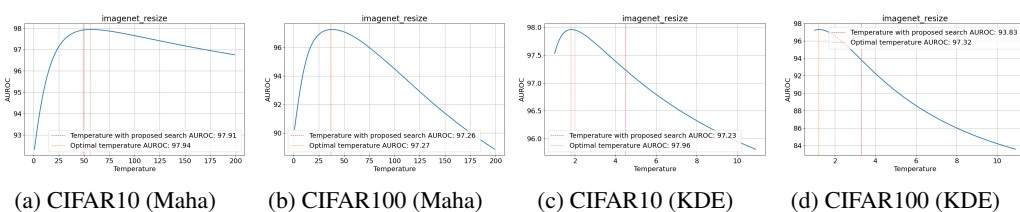

(a) CIFAR10 (Maha)  (b) CIFAR100 (Maha)  (c) CIFAR10 (KDE)  (d) CIFAR100 (KDE)

Figure 4: Plots of AUROC against temperatures. Imagenet-Resize is used as the OOD dataset.

for various datasets in Figure 4. In general, despite varying gaps in temperatures, the gap is small in terms of AUROC performance. We emphasize here again that the proposed method for temperature tuning only requires a small set of validation sets. In practice, the performance can further improve when OOD samples are available. Lastly, we note that, despite the seemingly worse performance of CDR_KDE on CIFARs when compared with CDR_Maha for experiments with the CIFAR datasets, the gap is largely due to the fact that the proposed temperature scaling method works better for CDR_Maha. With an optimal temperature, CDR_KDE performs on par with CDR_Maha. Additional details can be found in Appendix E.

**Additional Ablations** To facilitate further understanding of the proposed method, we have conducted additional experiments with different variants of the CDR. Details can be found in Appendix B.

## 6 DISCUSSIONS

We have proposed the conditional density ratio (CDR) for OOD detection. We have illustrated the importance of temperature scaling for our proposed method and devised a simple yet effective method to automatically tune the temperatures. Of the two variants of CDR scores tested, we have found CDR_KDE to be the more robust method across different models and tasks. Collectively, CDR provides us with an easy-to-implement, hyper-parameter-free method for outlier detection.

**Limitations** There are several potential limitations to the proposed method. Firstly, while the CDR score works well in most cases tested, the method can perform unsatisfactorily when either one of the conditional density approximations performs poorly. Secondly, we have made the simplistic assumption that the marginal distribution $p(z)$ is a good approximation to the true data distribution $p(x)$. This might not be the case in reality. Lastly, despite its effectiveness, the proposed method for temperature scaling may not be optimal in practice.

**Future Directions** We have just scratched the surface on density estimation-based approaches for OOD detection, and there are a lot of future directions for exploration. For instance, it can be beneficial to use a more expressive distribution like an exponential family (Peng et al., 2024) or even a non-parametric one to approximate $p(z|y)$. To take a step further, instead of modeling the distribution in latent representation space, we can instead leverage a generative model like normalizing flow to directly approximate $p(x)$ instead.

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

# A DERIVATIONS ON USING ID DENSITY FUNCTIONS FOR OUTLIER DETECTION

In this section, we formalize the intuition that, under the assumption that outlier samples have low likelihood under the inlier density function, a scoring function for outlier detection is optimal when based on the inlier density function. Derivations are adapted from (Sun et al., 2022).

Outlier detection can be formulated as a binary classification task of OOD and ID samples, where the OOD class is only available at test time. Suppose the test set $\{(z_i, g_i)\}$ is drawn *i.i.d.* from $P_{\mathbf{Z}, \mathbf{G}}$, where $z_i \in \mathbf{Z}$ is the feature embedding and $g_i \in \mathbf{G} = \{0 \text{ (OOD)}, 1 \text{ (ID)}\}$ is the corresponding label.

Let $P$ denote the marginal distribution of $\mathbf{Z}$. At test time, $P$ can be represented as:

$$P = \epsilon P_{ood} + (1 - \epsilon) P_{id},$$

where $P_{ood}$ and $P_{id}$ are the underlying distribution of $\mathbf{Z}$ for OOD and ID data, respectively, and $\epsilon$ represents the proportion of OOD samples at test time. The correspinding probability density function can be represented as $p_{ood}(z_i) = p(z_i | g_i = 0)$ and $p_{id}(z_i) = p(z_i | g_i = 1)$.

In general, OOD detection can be an ill-posed problem without assumptions on which out-distributions are relevant (Zhang et al., 2021). In this work, we make a reasonable assumption that OOD samples live in low density regions with respect to the inlier density function, and that OOD samples are uniformly distributed outside of the high-density region of ID data:

$$p_{ood}(z) = c_0 \mathbb{1}\{p_{id}(z) < c_1\},$$

for some constants $c_0$ and $c_1$.

Now, given the two distributions, the Bayes Optimal Classifier is a probabilistic model that makes the most probable prediction for a new example (Devroye et al., 2013). It is described using the Bayes Theorem, which provides a principled way for calculating a conditional probability, assuming the underlying density function is available.

$$h_{Bay}(z_i) = \mathbb{1}\{p(g_i = 1 | z_i) \geq \beta\}$$

From the above, the empirical estimation of the probability of $z$ being ID data $\hat{p}(g_i = 1 | z_i)$ can be expressed as:

$$p(g_i = 1 | z_i) = \frac{p(z_i | g_i = 1) \cdot p(g_i = 1)}{p(z_i)}$$

$$= \frac{p_{id}(z_i) \cdot p(g_i = 1)}{p_{id}(z_i) \cdot p(g_i = 1) + p_{ood}(z_i) \cdot p(g_i = 0)}$$

$$\hat{p}(g_i = 1 | z_i) = \frac{(1 - \epsilon)\, \hat{p}_{id}(z_i)}{(1 - \epsilon)\, \hat{p}_{id}(z_i) + \epsilon\, \hat{p}_{ood}(z_i)}$$

$$= \frac{(1 - \epsilon)\, \hat{p}_{id}(z_i)}{(1 - \epsilon)\hat{p}_{id}(z_i) + \epsilon\, \hat{c}_0 \mathbb{1}\{\hat{p}_{id}(z) < \hat{c}_1\}},$$

where $\hat{p}_{id}(z)$ and $\hat{p}_{ood}(z)$ denote estimates of the underlying densities $p_{id}(z)$ and $p_{ood}(z)$. Observe that, under our assumptions, the OOD detection problem boils down to accurately deriving the empirical estimation of $\hat{p}_{id}(z)$.

# B  ADDITIONAL ABLATION STUDY FOR CDR SCORE

In this section, we conduct additional ablation studies to further facilitate our understanding of our proposed method.

**How Effective is Aggregating CDR Scores Across Classes?**  We have shown in Equation 1 that the CDR score is an aggregation of class-specific CDR scores, and argued that such an approach gives us improved OOD detection performance. To empirically demonstrate the claim, we consider the following variants of CDR scores:

- "CDR_One": the variant where we only use the CDR score of the first class for OOD detection,

- "CDR_Max": the variant where we take the maximum of all class-conditional CDR scores for each sample for OOD detection,

- "CDR_Argmax": the variant where we take the class-conditional CDR score of the predicted class obtained from the pre-trained classifier model.

**Results**  We conduct experiments with the CDR_Maha score on the CIFAR datasets. Results are summarized in Table 5 and  6. As seen clearly, the relatively poor performances by "CDR_One", "CDR_Max" and "CDR_Argmax" highlight the importance of aggregating class-conditional CDR scores.

Table 5: OOD detection results on labeled CIFAR10 dataset with a pretrained DenseNet.

| OOD | Average | | SVHN | | LSUN-Resize | | Texture | | iSUN | |
|---|---|---|---|---|---|---|---|---|---|---|
| | FPR95↓ | AUROC ↑ | FPR95↓ | AUROC↑ | FPR95↓ | AUROC↑ | FPR95↓ | AUROC↑ | FPR95↓ | AUROC↑ |
| CDR | 10.53 | 97.72 | 11.87 | 97.75 | 5.37 | 98.8 | 18.42 | 95.7 | 6.47 | 98.64 |
| CDR_One | 38.54 | 92.52 | 27.11 | 95.36 | 40.28 | 94.03 | 48.97 | 86.47 | 37.8 | 94.21 |
| CDR_Max | 15.78 | 96.25 | 12.58 | 97.37 | 14.1 | 97.01 | 22.43 | 93.5 | 13.99 | 97.1 |
| CDR_ArgMax | 25.70 | 93.72 | 28.14 | 93.26 | 24.22 | 94.95 | 26.28 | 92.12 | 24.14 | 94.54 |

| Hard-OOD | Average | | ImageNet-Resize | | ImageNet-Fix | | LSUN-fix | | CIFAR100 | |
|---|---|---|---|---|---|---|---|---|---|---|
| | FPR95↓ | AUROC ↑ | FPR95↓ | AUROC↑ | FPR95↓ | AUROC↑ | FPR95↓ | AUROC↑ | FPR95↓ | AUROC↑ |
| CDR | 42.36 | 90.21 | 10.23 | 97.91 | 49.82 | 89.15 | 50.72 | 89.53 | 58.67 | 84.24 |
| CDR_One | 72.26 | 82.63 | 45.59 | 91.57 | 78.36 | 81.74 | 80.83 | 84.55 | 84.26 | 72.66 |
| CDR_Max | 50.02 | 86.23 | 15.79 | 96.29 | 57.63 | 84.7 | 61.95 | 84.27 | 64.69 | 79.66 |
| CDR_ArgMax | 66.28 | 76.59 | 25.16 | 93.85 | 77.04 | 72.25 | 81.29 | 73.23 | 81.62 | 67.02 |

Table 6: OOD detection results on labeled CIFAR100 dataset with a pretrained DenseNet.

| OOD | Average | | SVHN | | LSUN-Resize | | Texture | | iSUN | |
|---|---|---|---|---|---|---|---|---|---|---|
| | FPR95↓ | AUROC ↑ | FPR95↓ | AUROC↑ | FPR95↓ | AUROC↑ | FPR95↓ | AUROC↑ | FPR95↓ | AUROC↑ |
| CDR | 19.55 | 96.01 | 37.88 | 93.19 | 9.27 | 97.84 | 19.27 | 95.81 | 11.79 | 97.19 |
| CDR_Unlabeled | 18.82 | 96.12 | 35.98 | 93.53 | 9.09 | 97.87 | 18.78 | 95.86 | 11.43 | 97.23 |
| CDR_One | 35.31 | 92.65 | 47.01 | 91.91 | 31.39 | 93.07 | 31.15 | 93.15 | 31.67 | 92.46 |
| CDR_Max | 23.83 | 95.12 | 45.82 | 91.41 | 11.83 | 97.46 | 23.42 | 94.82 | 14.26 | 96.78 |
| CDR_ArgMax | 42.13 | 87.45 | 69.86 | 78.85 | 31.36 | 91.91 | 34.5 | 88.1 | 32.79 | 90.95 |

| Hard-OOD | Average | | ImageNet-Resize | | ImageNet-Fix | | LSUN-fix | | CIFAR100 | |
|---|---|---|---|---|---|---|---|---|---|---|
| | FPR95↓ | AUROC ↑ | FPR95↓ | AUROC↑ | FPR95↓ | AUROC↑ | FPR95↓ | AUROC↑ | FPR95↓ | AUROC↑ |
| CDR | 71.33 | 75.90 | 12.34 | 97.26 | 84.51 | 74.15 | 93.06 | 65.66 | 95.42 | 66.52 |
| CDR_Unlabeled | 71.21 | 76.07 | 12.33 | 97.24 | 84.16 | 74.38 | 92.99 | 65.98 | 95.34 | 66.68 |
| CDR_One | 78.32 | 67.57 | 27.68 | 94.01 | 89.06 | 67.21 | 98.56 | 51.66 | 97.96 | 57.38 |
| CDR_Max | 72.52 | 74.18 | 15.89 | 96.64 | 84.76 | 73.03 | 94.82 | 60.39 | 94.6 | 66.64 |
| CDR_ArgMax | 80.51 | 54.83 | 30.84 | 90.6 | 94.12 | 46.52 | 97.92 | 46.46 | 99.15 | 35.74 |

# C  EXPERIMENTS WITH BERT ON NLP DATASETS

We conduct additional experiments with NLP datasets using the Amazon (He & McAuley, 2016), Imdb (Maas et al., 2011), Yelp (Zhang et al., 2015), 20newsgroups (Lang, 1995), and Agnews (Zhang et al., 2015) datasets. For simplicity, we use the datasets curated by ADBench (Han et al., 2022) instead of the raw datasets. The datasets involve embedding vectors extracted using a BERT (Devlin et al., 2018) model pretrained on the BookCorpus and English Wikipedia. The datasets are subsampled and only contain 10,000 samples each. More details on the datasets can be found at (Han et al., 2022). For experimental purposes, we consider the following scenarios for supervised model training and outlier detection:

- The 20 newsgroups dataset contains six different classes, respectively. We treat the samples from the first three classes as inliers. To simulate a hard OOD sample scenario, we treat the remaining classes as outliers.

- The Agnews dataset contains four different classes, respectively. We treat the samples from the first three classes as inliers. Similarly, we treat the remaining one as the outlier.

In addition to near-OOD samples, we also consider data from Amazon, Imdb, and Yelp as outliers for each case. During training, a linear classifier is trained to distinguish between the inlier classes. We also compute the mean and precision matrices using the inlier classes. We use 60% of the available inlier data for training and the remaining samples for testing. We hold out 10% of the samples in the training data as a validation set and use it to estimate the mean, covariance matrix, and temperature parameters for Mahalanobis, GEM, and CDR score. For KNN, since no OOD samples are available for tuning the hyperparameter, we set $k = 10$ for our experiments. Results are summarized in Table 7 and 8. Similar to other results, the proposed CDR loss yields competitive performance in most of the cases tested.

Table 7: OOD detection results using models trained with the first 3 classes of 20newsgroups dataset as inlier samples. **20news Outlier** corresponds to the remaining 3 classes in the dataset.

| | **20news Outlier** | | **Agnew** | | **Amazon** | | **Imdb** | | **Yelp** | |
| | FPR95 | AUROC | FPR95 | AUROC | FPR95 | AUROC | FPR95 | AUROC | FPR95 | AUROC |
|---|---|---|---|---|---|---|---|---|---|---|
| MSP | 93.64 | 69.33 | 95.38 | 56.34 | 91.05 | 72.44 | 96.60 | 72.83 | 89.76 | 75.02 |
| Energy | 93.31 | 66.14 | 95.01 | 55.29 | 84.04 | 76.51 | 95.95 | 66.44 | 78.38 | 81.23 |
| Maha | 82.25 | 70.67 | 5.94 | 98.69 | 46.98 | 91.63 | 36.92 | 94.47 | 54.87 | 90.55 |
| GEM | 82.25 | 70.65 | **5.93** | **98.69** | 46.98 | 91.62 | 36.93 | 94.46 | 54.87 | 90.54 |
| KNN | 88.50 | 65.51 | 22.24 | 91.84 | 68.54 | 87.40 | 66.15 | 91.11 | 74.35 | 85.27 |
| CDR_KDE | 78.35 | 76.21 | 29.61 | 88.59 | 34.45 | 91.89 | 45.82 | 90.21 | **32.18** | 92.78 |
| CDR_Maha | **77.88** | **76.89** | 7.79 | 98.28 | **32.35** | **94.35** | 28.58 | **95.42** | 35.59 | **94.08** |

Table 8: OOD detection results using models trained with the first 3 classes of Agnews dataset as inlier samples. **Agnews Outlier** corresponds to the remaining class in the dataset.

| | **20news** | | **Agnews Outlier** | | **Amazon** | | **Imdb** | | **Yelp** | |
| | FPR95 | AUROC | FPR95 | AUROC | FPR95 | AUROC | FPR95 | AUROC | FPR95 | AUROC |
|---|---|---|---|---|---|---|---|---|---|---|
| MSP | 54.93 | 88.00 | 83.45 | 73.27 | 53.85 | 91.40 | 69.54 | 87.16 | 46.18 | 92.75 |
| Energy | 34.92 | 89.99 | 78.93 | 74.70 | 30.06 | 94.56 | 66.78 | 87.38 | 23.19 | 95.63 |
| Maha | 24.23 | 93.78 | 73.98 | 75.80 | 19.57 | 96.61 | 16.15 | 97.27 | 14.26 | 97.10 |
| GEM | 24.23 | 93.77 | 73.99 | 75.79 | 19.61 | 96.61 | 16.17 | 97.26 | 14.28 | 97.09 |
| KNN | 51.83 | 86.12 | 88.31 | 68.24 | 71.64 | 85.34 | 76.00 | 86.23 | 57.14 | 89.04 |
| CDR_KDE | 13.92 | 96.67 | 71.39 | 77.71 | 4.83 | 98.80 | 9.56 | 98.05 | 3.63 | 99.12 |
| CDR_Maha | **12.26** | **97.13** | **68.83** | **79.07** | **3.88** | **98.88** | **5.78** | **98.55** | **2.88** | **99.17** |

# D  ADDITIONAL EXPERIMENTS ON IMAGENET

We provide additional experimental data on models trained with the ImageNet dataset. We empirically observed that, given the same sets of ID datasets and OOD datasets, the OOD detection performances vary widely across different training strategies and model architectures for the ImageNet dataset. Even when the architecture is identical, detection performances across different methods fluctuate significantly. To holistically benchmark different strategies for OOD detection, we conduct experiments using the following architectures and weights provided by PyTorch:

- `Wide_ResNet101_2_Weights.IMAGENET1K_V1`,
- `ResNet50_Weights.IMAGENET1K_V1`,
- `ResNet50_Weights.IMAGENET1K_V2`,
- `ViT_B_16_Weights.IMAGENET1K_V1`,
- `DenseNet121_Weights.IMAGENET1K_V1`,
- `MobileNet_V2_Weights.IMAGENET1K_V1`.

Details on model training can be found at `https://pytorch.org/vision/stable/models.html`.

Model-wise performances are summarized in Tables 9, 10, 11 12, 13 and 14. Observe that, the relative OOD detection performances for different strategies can vary widely across different model architectures and datasets for the ImageNet dataset. This is the case even when model architectures stay the same, as exemplified by the comparison between Table 10 and 11. We leave it to future work to investigate the impact of model architectures and training configurations on the performance of OOD detection.

To better compare the overall strategies, we compute the mean and standard deviation across different model architectures for all OOD datasets. Results are summarized in Figure 5. We highlight that our proposed CDR_KDE not only performs the best among all the methods in most scenarios, but the variances in performances across different models are also one of the smallest, making it an ideal method as an easy-to-implement, hyper-parameter-free benchmark method for the important task of outlier detection.

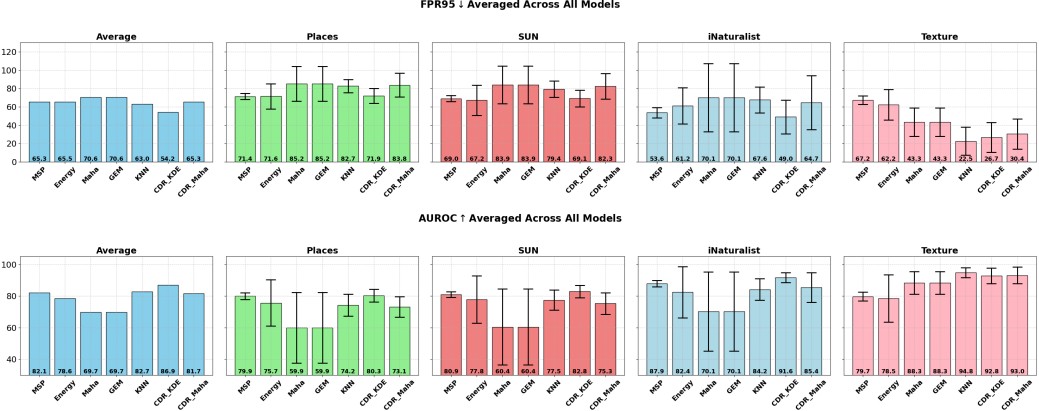

Figure 5: Bar plots of AUROCs and FPR95s for different OOD datasets averaged across all the models tested. In general, the proposed CDR_KDE not only achieves superior performances for most of the cases, but also exhibits lower variances across different models.

Table 9: OOD detection results on ImageNet dataset with a pretrained Wide-ResNet50.

| | Average | | Places | | SUN | | iNaturalist | | Texture | |
|---|---|---|---|---|---|---|---|---|---|---|
| | FPR95↓ | AUROC↑ | FPR95↓ | AUROC↑ | FPR95↓ | AUROC↑ | FPR95↓ | AUROC↑ | FPR95↓ | AUROC↑ |
| MSP | 61.07 | 84.29 | 67.75 | 82.41 | 64.97 | 83.5 | 46.58 | 90.04 | 64.98 | 81.21 |
| Energy | 60.55 | 86.31 | 66.12 | 84.32 | 61.26 | 86.21 | 55.73 | 90.49 | 59.08 | 84.20 |
| Maha | 79.52 | 74.83 | 94.4 | 65.87 | 92.97 | 67.33 | 87.74 | 75.42 | 42.98 | 90.68 |
| GEM | 79.52 | 74.83 | 94.4 | 65.87 | 92.97 | 67.33 | 87.74 | 75.42 | 42.98 | 90.68 |
| KNN | 52.22 | 87.24 | 76.42 | 78.9 | 71.39 | 82.26 | 47.49 | 91.32 | 13.58 | 96.47 |
| CDR_KDE | 45.61 | 91.02 | 63.22 | 86.07 | 59.36 | 88.39 | 43.74 | 93.47 | 16.13 | 96.15 |
| CDR_Maha | 64.82 | 85.61 | 83.66 | 78.62 | 81.9 | 81.08 | 71.16 | 86.74 | 22.55 | 96.00 |

Table 10: OOD detection results on ImageNet dataset with a pretrained ResNet50 with `IMAGENET1K_V1` weights.

| | Average | | Places | | SUN | | iNaturalist | | Texture | |
|---|---|---|---|---|---|---|---|---|---|---|
| | FPR95↓ | AUROC ↑ | FPR95↓ | AUROC↑ | FPR95↓ | AUROC↑ | FPR95↓ | AUROC↑ | FPR95↓ | AUROC↑ |
| MSP | 65.26 | 82.89 | 72.19 | 80.7 | 69.33 | 81.79 | 53.02 | 88.47 | 66.51 | 80.58 |
| Energy | 57.48 | 87.07 | 65.73 | 84.1 | 58.65 | 86.68 | 53.39 | 90.69 | 52.16 | 86.81 |
| Maha | 83.08 | 64.51 | 97.5 | 52.72 | 97.47 | 53.12 | 94.38 | 62.02 | 42.96 | 90.19 |
| GEM | 83.08 | 64.51 | 97.5 | 52.72 | 97.47 | 53.12 | 94.38 | 62.01 | 42.96 | 90.19 |
| KNN | 53.57 | 85.87 | 74.46 | 78.14 | 66.58 | 82.69 | 61.58 | 85.58 | 11.67 | 97.08 |
| CDR_KDE | 53.49 | 88.75 | 72.8 | 82.6 | 67.87 | 85.46 | 58.07 | 90.77 | 15.23 | 96.15 |
| CDR_Maha | 69.22 | 82.13 | 90.29 | 73.15 | 88.57 | 75.84 | 81.44 | 82.6 | 16.58 | 96.94 |

Table 11: OOD detection results on ImageNet dataset with a pretrained ResNet50 with `IMAGENET1K_V2` weights.

| | Average | | Places | | SUN | | iNaturalist | | Texture | |
|---|---|---|---|---|---|---|---|---|---|---|
| | FPR95↓ | AUROC ↑ | FPR95↓ | AUROC↑ | FPR95↓ | AUROC↑ | FPR95↓ | AUROC↑ | FPR95↓ | AUROC↑ |
| MSP | 70.14 | 78.69 | 73.87 | 76.7 | 71.4 | 78.71 | 61.67 | 84.48 | 73.6 | 74.88 |
| Energy | 97.28 | 49.63 | 97.67 | 48.92 | 97.8 | 50.13 | 98.06 | 50.9 | 95.59 | 48.56 |
| Maha | 45.07 | 89.05 | 65.59 | 82.96 | 60.88 | 85.48 | 26.84 | 94.98 | 26.95 | 92.77 |
| GEM | 45.07 | 89.05 | 65.59 | 82.96 | 60.88 | 85.48 | 26.84 | 94.98 | 26.95 | 92.77 |
| KNN | 68.05 | 82.75 | 85.34 | 76.4 | 84.98 | 77.96 | 74.74 | 84.18 | 27.15 | 92.47 |
| CDR_KDE | 52.43 | 83.55 | 70.62 | 74.38 | 67.16 | 77.85 | 34.67 | 92.74 | 37.25 | 89.21 |
| CDR_Maha | 54.52 | 83.66 | 75.36 | 73.58 | 71.92 | 76.81 | 37.48 | 92.83 | 33.32 | 91.41 |

Table 12: OOD detection results on ImageNet dataset with a pretrained DenseNet121.

| | Average | | Places | | SUN | | iNaturalist | | Texture | |
|---|---|---|---|---|---|---|---|---|---|---|
| | FPR95↓ | AUROC ↑ | FPR95↓ | AUROC↑ | FPR95↓ | AUROC↑ | FPR95↓ | AUROC↑ | FPR95↓ | AUROC↑ |
| MSP | 63.66 | 82.75 | 69.77 | 81.07 | 67.77 | 81.52 | 49.75 | 89.09 | 67.34 | 79.31 |
| Energy | 51.16 | 87.58 | 58.78 | 85.04 | 53.05 | 87.25 | 40.28 | 92.63 | 52.54 | 85.41 |
| Maha | 79.43 | 62.26 | 98.01 | 45.47 | 98.16 | 46.08 | 94.42 | 63.69 | 27.11 | 93.81 |
| GEM | 79.43 | 62.26 | 98.01 | 45.46 | 98.16 | 46.07 | 94.42 | 63.69 | 27.11 | 93.81 |
| KNN | 66.98 | 79.68 | 88.33 | 68.75 | 86.23 | 72.81 | 78.2 | 80.42 | 15.14 | 96.74 |
| CDR_KDE | 53.32 | 88.31 | 76.29 | 81.2 | 73.4 | 83.81 | 49.36 | 91.8 | 14.22 | 96.41 |
| CDR_Maha | 70.92 | 81.90 | 93.92 | 71.3 | 93.39 | 73.94 | 81.82 | 85.04 | 14.56 | 97.32 |

Table 13: OOD detection results on ImageNet dataset with a pretrained MobileNet_V2.

| | Average | | Places | | SUN | | iNaturalist | | Texture | |
|---|---|---|---|---|---|---|---|---|---|---|
| | FPR95↓ | AUROC ↑ | FPR95↓ | AUROC↑ | FPR95↓ | AUROC↑ | FPR95↓ | AUROC↑ | FPR95↓ | AUROC↑ |
| MSP | 70.16 | 80.81 | 76.44 | 78.29 | 74.06 | 79 | 59.31 | 86.82 | 70.83 | 79.11 |
| Energy | 58.77 | 86.63 | 66.2 | 83.23 | 59.31 | 86.23 | 55.08 | 90.42 | 54.5 | 86.63 |
| Maha | 91.59 | 38.79 | 99.29 | 27.6 | 99.55 | 24.41 | 99.38 | 28.73 | 68.12 | 74.43 |
| GEM | 91.59 | 38.79 | 99.29 | 27.59 | 99.55 | 24.41 | 99.38 | 28.72 | 68.12 | 74.43 |
| KNN | 70.40 | 74.86 | 92.28 | 62.66 | 88.54 | 67.07 | 85.33 | 73.05 | 15.46 | 96.66 |
| CDR_KDE | 68.00 | 84.72 | 85.02 | 77.92 | 84.7 | 80.5 | 80.06 | 85.82 | 22.23 | 94.64 |
| CDR_Maha | 81.87 | 71.70 | 96.86 | 61.89 | 97.61 | 62.66 | 96.32 | 69.04 | 36.67 | 93.22 |

Table 14: OOD detection results on ImageNet dataset with a pretrained ViT_B_16.

| | Average | | Places | | SUN | | iNaturalist | | Texture | |
|---|---|---|---|---|---|---|---|---|---|---|
| | FPR95↓ | AUROC ↑ | FPR95↓ | AUROC↑ | FPR95↓ | AUROC↑ | FPR95↓ | AUROC↑ | FPR95↓ | AUROC↑ |
| MSP | 61.58 | 83.21 | 68.35 | 80.49 | 66.46 | 81 | 51.24 | 88.26 | 60.25 | 83.07 |
| Energy | 67.94 | 74.24 | 74.81 | 68.37 | 73.21 | 70.14 | 64.64 | 79.23 | 59.11 | 79.2 |
| Maha | 45.13 | 88.63 | 56.48 | 84.68 | 54.58 | 86.02 | 18.04 | 95.98 | 51.42 | 87.84 |
| GEM | 45.13 | 88.63 | 56.48 | 84.68 | 54.58 | 86.02 | 18.04 | 95.98 | 51.42 | 87.84 |
| KNN | 66.98 | 85.54 | 79.16 | 80.24 | 78.46 | 81.95 | 58.51 | 90.37 | 51.77 | 89.59 |
| CDR_KDE | 52.30 | 84.99 | 63.68 | 79.68 | 62.06 | 80.97 | 28.25 | 94.75 | 55.21 | 84.56 |
| CDR_Maha | 50.53 | 85.18 | 62.42 | 80.15 | 60.59 | 81.25 | 20.14 | 95.9 | 58.95 | 83.42 |

# E ADDITIONAL DETAILS ON THE ABLATION STUDY OF TEMPERATURE SCALING

In this section, we provide additional details on the ablation study on temperature scaling. Specifically, we investigate how temperature scaling affects benchmark methods like MSP and energy. To this end, we apply the temperatures found using the algorithm proposed for temperature search in this paper to all the relevant benchmark methods using models trained on the CIFAR10 and CIFAR100 datasets. Results are summarized in Table 15. Interestingly, we see that the proposed temperature scaling leads to better performance for MSP but degrades the OOD detection effectiveness of energy and GEM. Next, we provide in Figures 6, 7, 8 and 9 the plots of AUROC against temperatures

Table 15: Average OOD detection results **with/without temperature scaling**. We compare each pair of the results, and boldface the better results.

| | CIFAR10 | | | | CIFAR100 | | | |
| | Average OOD Datasets | | Average Hard OOD Dataset | | Average OOD Datasets | | Average Hard OOD Datasets | |
| | FPR95↓ | AUROC↑ | FPR95↓ | AUROC ↑ | FPR95↓ | AUROC ↑ | FPR95↓ | AUROC ↑ |
|---|---|---|---|---|---|---|---|---|
| MSP | **50.10** / 57.61 | **91.61** / 90.62 | **58.25** / 63.89 | **88.78** / 88.00 | **81.74** / 84.41 | **76.16** / 73.71 | **83.97** / 85.73 | **74.07** / 72.39 |
| Energy | 37.28 / **37.26** | 91.88 / **91.99** | 48.41 / **47.86** | 88.73 / **88.88** | 83.84 / **82.09** | 76.82 / **77.20** | 84.89 / **84.05** | 73.55 / **74.06** |
| GEM | 27.76 / **20.38** | 92.63 / **95.58** | 68.60 / **61.58** | 73.01 / **81.81** | 46.15 / **35.65** | 86.69 / **91.67** | 81.73 / **78.47** | 52.96 / **62.19** |
| CDR_KDE | 14.68 / **11.41** | 97.01 / **97.64** | 38.65 / **38.22** | 91.05 / **91.14** | 30.52 / **25.89** | 92.69 / **94.43** | 69.87 / **68.98** | 77.91 / **78.22** |
| CDR_Maha | **10.53** / 27.76 | **97.72** / 92.52 | **42.36** / 68.81 | **90.21** / 72.47 | **19.55** / 44.84 | **96.01** / 87.71 | **71.33** / 81.33 | **75.90** / 54.15 |

on models trained using CIFAR10 and CIFAR100 for all the OOD datasets for both the CDR_KDE and CDR_Maha. Similar to the observation from Figure 4, temperature scaling is beneficial for CDR_Maha. Though it can degrade the performance of CDR_KDE in some cases, the amount of degradation is very minimal in most cases when temperature scaling is harmful. Lastly, we note that, though CDR_Maha is shown as the better method when compared with CDR_KDE in Table 1 and 2, the improvement comes largely from a better selected temperature using the proposed method. When comparing the optimal temperature AUROC, CDR_KDE performs on par with CDR_Maha in nearly all of the cases, as can be seen from Figures 6, 7, 8 and 9. However, according to CDR_Maha, CDR_KDE is much less sensitive to the choice of temperature, as can be seen from the much smaller variations in AUROCs when temperatures increase in the figures.

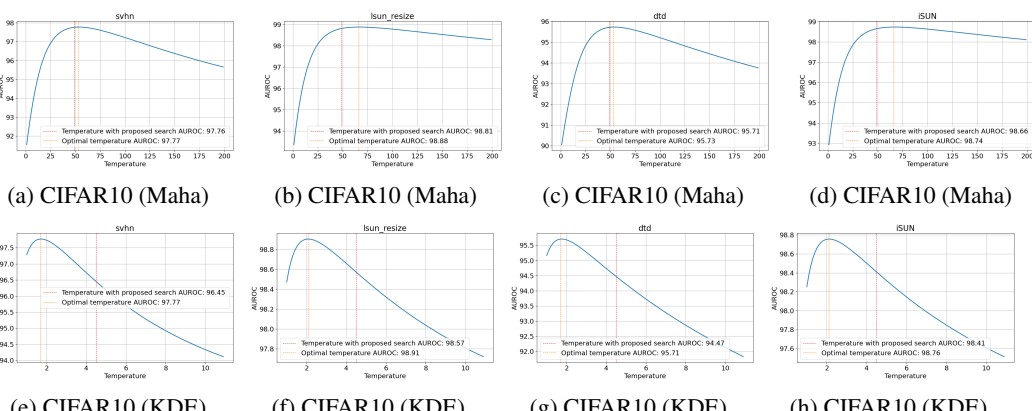

| (a) CIFAR10 (Maha) | (b) CIFAR10 (Maha) | (c) CIFAR10 (Maha) | (d) CIFAR10 (Maha) |
|---|---|---|---|
| (e) CIFAR10 (KDE) | (f) CIFAR10 (KDE) | (g) CIFAR10 (KDE) | (h) CIFAR10 (KDE) |

Figure 6: Plots of AUROC against temperatures for different methods and datasets.

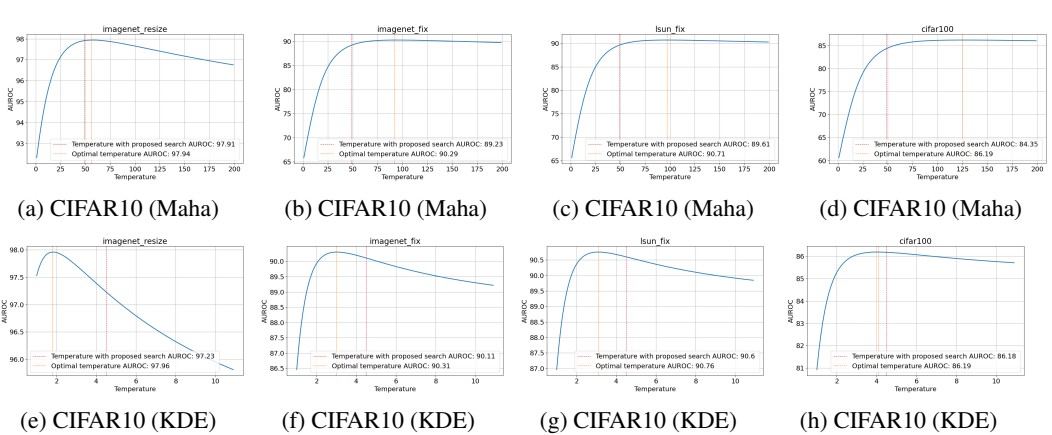

Figure 7: Plots of AUROC against temperatures for different methods and datasets.

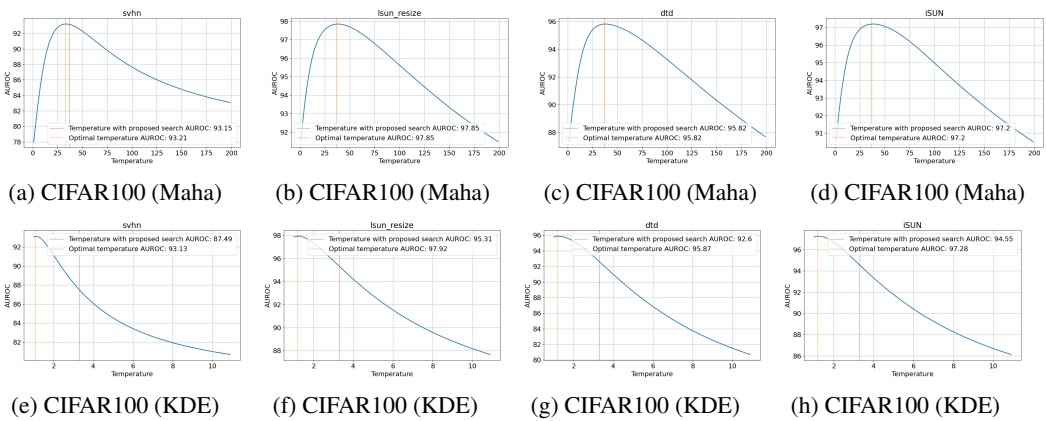

Figure 8: Plots of AUROC against temperatures for different methods and datasets.

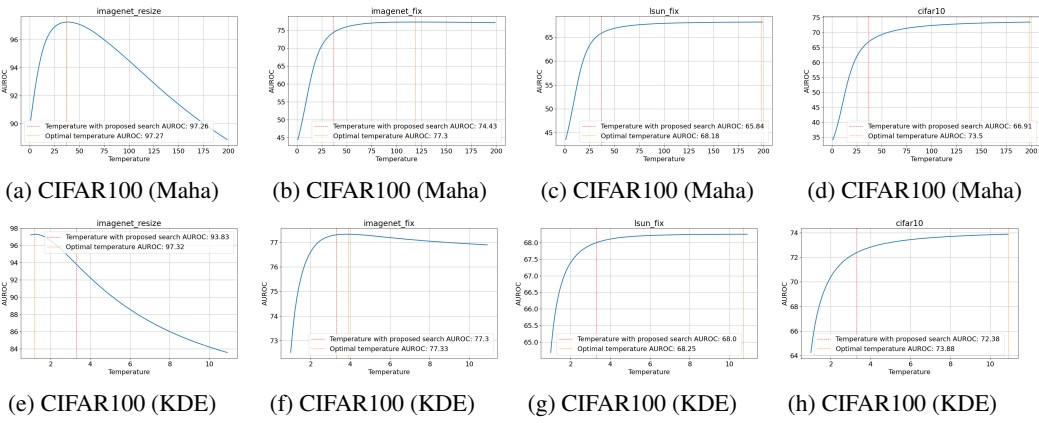

Figure 9: Plots of AUROC against temperatures for different methods and datasets.

# F    EXPERIMENTS WITH RESNET ON CIFAR

In light of the variance in OOD detection performances of different models on the ImageNet dataset, we conduct additional experiments on OOD performance with a ResNet34 model. We use the identical training setup as the DenseNet model. Results are summarized in Table 16 and 17. In general, the impact of model architecture choices is smaller on CIFAR experiments, and the same conclusions hold for experiments with the ResNet34 models.

Table 16: OOD detection results on labeled CIFAR10 dataset with a pretrained ResNet.

| OOD | Average | | SVHN | | LSUN-Resize | | Texture | | iSUN | |
|---|---|---|---|---|---|---|---|---|---|---|
| | FPR95↓ | AUROC↑ | FPR95↓ | AUROC↑ | FPR95↓ | AUROC↑ | FPR95↓ | AUROC↑ | FPR95↓ | AUROC↑ |
| GradNorm | 82.04 | 55.11 | 86.34 | 50.78 | 77.93 | 61.07 | 82.55 | 50.3 | 81.34 | 58.3 |
| MSP | 48.18 | 87.33 | 54.85 | 87.83 | 41.61 | 89.72 | 49.17 | 85.75 | 47.09 | 86.03 |
| Energy | 47.18 | 87.40 | 53.86 | 87.87 | 40.43 | 89.83 | 48.37 | 85.77 | 46.05 | 86.12 |
| Maha | 49.69 | 89.67 | 31.71 | 93.76 | 64.72 | 87.04 | 35.2 | 91.93 | 67.12 | 85.96 |
| GEM | 49.70 | 89.67 | 31.73 | 93.75 | 64.74 | 87.04 | 35.21 | 91.92 | 67.12 | 85.96 |
| KNN | 41.05 | 89.02 | 46.45 | 88.94 | 34.84 | 91.31 | 41.21 | 87.73 | 41.68 | 88.1 |
| CDR_KDE | 38.96 | 90.34 | 41.45 | 91.13 | 35.46 | 91.68 | 37.52 | 89.83 | 41.41 | 88.71 |
| CDR_Maha | **33.08** | **92.52** | **29.36** | **93.8** | **34.04** | **92.84** | **29.06** | **92.56** | **39.87** | **90.87** |

| Hard-OOD | Average | | ImageNet-Resize | | ImageNet-Fix | | LSUN-fix | | CIFAR100 | |
|---|---|---|---|---|---|---|---|---|---|---|
| | FPR95↓ | AUROC↑ | FPR95↓ | AUROC↑ | FPR95↓ | AUROC↑ | FPR95↓ | AUROC↑ | FPR95↓ | AUROC↑ |
| GradNorm | 86.11 | 46.71 | 81.66 | 60.19 | 86.71 | 42.97 | 89.79 | 37.53 | 86.28 | 46.13 |
| MSP | 52.83 | 84.39 | 55.27 | 81.87 | 50.18 | 85.85 | 50.19 | 86.52 | 55.68 | 83.33 |
| Energy | 52.03 | 84.44 | 54.34 | 81.94 | 49.37 | 85.9 | 49.37 | 86.56 | 55.05 | 83.37 |
| Maha | 71.96 | 82.51 | 67.85 | 83.34 | 72.65 | 82.97 | 72.54 | 84.02 | 74.81 | 79.71 |
| GEM | 71.98 | 82.51 | 67.88 | 83.34 | 72.66 | 82.96 | 72.55 | 84.02 | 74.81 | 79.71 |
| KNN | 50.28 | 84.61 | 52.22 | 83.49 | 47.23 | 85.9 | 47.56 | 86.52 | 54.12 | 82.51 |
| CDR_KDE | **49.21** | 86.32 | 50.81 | 84.79 | **46.33** | 87.53 | **46.77** | 88.26 | **52.91** | 84.7 |
| CDR_Maha | 49.27 | **87.67** | **48.51** | **87.51** | 47.27 | **88.5** | 47.23 | **89.42** | 54.06 | **85.26** |

Table 17: OOD detection results on labeled CIFAR100 dataset with a pretrained ResNet.

| OOD | Average | | SVHN | | LSUN-Resize | | Texture | | iSUN | |
|---|---|---|---|---|---|---|---|---|---|---|
| | FPR95↓ | AUROC↑ | FPR95↓ | AUROC↑ | FPR95↓ | AUROC↑ | FPR95↓ | AUROC↑ | FPR95↓ | AUROC↑ |
| GradNorm | 74.83 | 70.88 | 58.36 | 84.97 | 78.22 | 64.15 | 83.24 | 67.92 | 79.49 | 66.46 |
| MSP | 77.92 | 80.44 | 77.26 | 82.51 | 72.65 | 82.39 | 86.06 | 75.56 | 75.69 | 81.29 |
| Energy | 76.62 | 80.76 | 76.39 | 82.89 | 69.86 | 82.96 | 86.72 | 75.39 | 73.51 | 81.81 |
| Maha | 84.68 | 69.67 | 87.25 | 70.72 | 92.67 | 64.97 | 65.69 | 79.3 | 93.11 | 63.7 |
| GEM | 84.68 | 69.67 | 87.25 | 70.71 | 92.68 | 64.96 | 65.69 | 79.3 | 93.11 | 63.69 |
| KNN | 67.71 | 83.99 | 64.99 | 86.37 | 61.53 | 85.94 | 78.62 | 79.04 | 65.7 | 84.61 |
| CDR_KDE | **61.07** | 85.30 | **57.28** | 87.13 | **59.36** | **86.31** | 64.02 | 82.8 | **63.6** | **84.94** |
| CDR_Maha | 63.52 | **86.54** | 62.42 | **87.34** | 69.51 | 85.98 | **49.43** | **88.36** | 72.7 | 84.48 |

| Hard-OOD | Average | | ImageNet-Resize | | ImageNet-Fix | | LSUN-fix | | CIFAR10 | |
|---|---|---|---|---|---|---|---|---|---|---|
| | FPR95↓ | AUROC↑ | FPR95↓ | AUROC↑ | FPR95↓ | AUROC↑ | FPR95↓ | AUROC↑ | FPR95↓ | AUROC↑ |
| GradNorm | 86.25 | 62.02 | 81.05 | 64.3 | 85.84 | 61.92 | 92.61 | 56.36 | 85.48 | 65.5 |
| MSP | 81.72 | 77.15 | 77.62 | 80.46 | 79.44 | 78.11 | **89.61** | **71.33** | **80.2** | **78.69** |
| Energy | 81.05 | 77.19 | 75.19 | 80.92 | 78.96 | 78.09 | 89.79 | 71.08 | 80.24 | 78.67 |
| Maha | 95.43 | 55.87 | 91.48 | 65.45 | 95.76 | 54.81 | 96.97 | 52.7 | 97.49 | 50.5 |
| GEM | 95.43 | 55.85 | 91.48 | 65.43 | 95.76 | 54.79 | 96.97 | 52.69 | 97.5 | 50.49 |
| KNN | 80.17 | 77.31 | 68.66 | 83.8 | 80.29 | 77.72 | 90.39 | 69.56 | 81.33 | 78.14 |
| CDR_KDE | **79.10** | **77.84** | **64.21** | 84.63 | **79.94** | **78.15** | 89.99 | 70.31 | 82.27 | 78.28 |
| CDR_Maha | 84.30 | 75.29 | 69.48 | 85.22 | 86.49 | 74.56 | 91.91 | 67.44 | 89.31 | 73.92 |

# G    ADDITIONAL DETAILS ON SELF-SUPERVISED OUTLIER DETECTION TASK

To showcase the general applicability of the proposed method, we apply the CDR score to the scenario of anomaly detection for self-supervised learning. Specifically, we train a DenseNet121 model using an identical setup as a supervised scenario. However, instead of the underlying labels, for each image, we randomly apply a rotation from the set of $\{0°, 90°, 180°, 270°\}$ to each image and have the neural network predict the angle of rotation of the image. At test time, we find the mean vectors of each rotation and the covariance matrix given a small set of validation samples. Scoring functions for outlier detection can then be applied, given an embedding vector. During inference, we apply all rotations to each image. The overall anomaly scores are obtained by taking the average across all rotations. We provide in Table 18 and 19 the full breakdown of the aggregated results in Table 4.

Table 18: OOD detection results on self-supervised CIFAR10 dataset with a pretrained DenseNet.

| OOD | Average | | SVHN | | LSUN-Resize | | Texture | | iSUN | |
|---|---|---|---|---|---|---|---|---|---|---|
| | FPR95↓ | AUROC ↑ | FPR95↓ | AUROC↑ | FPR95↓ | AUROC↑ | FPR95↓ | AUROC↑ | FPR95↓ | AUROC↑ |
| GradNorm | 44.60 | 90.69 | 62.82 | 85.51 | 41.77 | 91.7 | 29.79 | 94.77 | 44.01 | 90.77 |
| MSP | 50.87 | 91.07 | 48.83 | 92.53 | 57.05 | 89.09 | 39.7 | 93.85 | 57.89 | 88.82 |
| Energy | 41.10 | 92.63 | 37.19 | 94.06 | 46.3 | 91.08 | 32.23 | 94.91 | 48.68 | 90.48 |
| Maha | 46.84 | 87.45 | 68.95 | 80.9 | 39.36 | 91.2 | 42.39 | 86.91 | 36.64 | 90.79 |
| GEM | 46.94 | 87.39 | 69.07 | 80.8 | 39.46 | 91.17 | 42.52 | 86.84 | 36.72 | 90.75 |
| KNN | 59.03 | 88.32 | 42.69 | 92.86 | 77.01 | 83.67 | 41.01 | 93.19 | 75.41 | 83.57 |
| CDR_KDE | 32.55 | 94.19 | 36.89 | 94.19 | 34.82 | 93.31 | 22.85 | 96.39 | 35.63 | 92.85 |
| CDR_Maha | **16.59** | **96.83** | **19.36** | **96.62** | **16.11** | **96.69** | **13.23** | **97.85** | **17.64** | **96.15** |

| Hard-OOD | Average | | ImageNet-Resize | | ImageNet-Fix | | LSUN-fix | | CIFAR100 | |
|---|---|---|---|---|---|---|---|---|---|---|
| | FPR95↓ | AUROC ↑ | FPR95↓ | AUROC↑ | FPR95↓ | AUROC↑ | FPR95↓ | AUROC↑ | FPR95↓ | AUROC↑ |
| GradNorm | 64.76 | 81.82 | 36.64 | 92.75 | 68.26 | 81.26 | 78.7 | 78.37 | 75.43 | 74.89 |
| MSP | 68.65 | 82.82 | 52.86 | 90.65 | 69.81 | 82.73 | 75.03 | 80.36 | 76.88 | 77.55 |
| Energy | 62.56 | 84.35 | 43.74 | 92.06 | 63.36 | 84.51 | 69.48 | 83.01 | 73.66 | 77.81 |
| Maha | 79.77 | 61.22 | 31.97 | 92.17 | 94.41 | 54.38 | 98.31 | 43.34 | 94.39 | 54.97 |
| GEM | 79.82 | 61.13 | 32.08 | 92.14 | 94.46 | 54.27 | 98.31 | 43.24 | 94.44 | 54.88 |
| KNN | 74.29 | 78.93 | 73.76 | 85.83 | 72.04 | 79.33 | 71.23 | 75.42 | 80.12 | 75.13 |
| CDR_KDE | 58.83 | **85.12** | 31.2 | 94.31 | 62.12 | **84.84** | 68.79 | **83.32** | 73.22 | **78.01** |
| CDR_Maha | **58.09** | 84.84 | **14.36** | **97.15** | 65.25 | 83.83 | 79 | 80.73 | 73.76 | 77.65 |

Table 19: OOD detection results on self-supervised CIFAR100 dataset with a pretrained DenseNet.

| OOD | Average | | SVHN | | LSUN-Resize | | Texture | | iSUN | |
|---|---|---|---|---|---|---|---|---|---|---|
| | FPR95↓ | AUROC ↑ | FPR95↓ | AUROC↑ | FPR95↓ | AUROC↑ | FPR95↓ | AUROC↑ | FPR95↓ | AUROC↑ |
| GradNorm | 91.30 | 65.41 | 92.89 | 69.95 | 97.51 | 57.83 | 76.74 | 77.4 | 98.04 | 56.45 |
| MSP | 87.85 | 71.54 | 86.32 | 75.33 | 92.96 | 66.71 | 78.63 | 77.71 | 93.5 | 66.41 |
| Energy | 87.51 | 72.30 | 85.06 | 77.61 | 93.45 | 66.82 | 77.54 | 78.46 | 93.98 | 66.29 |
| Maha | 53.87 | 80.98 | 89.04 | 68.72 | 22.83 | 94.73 | 76.31 | 67.49 | 27.29 | 92.96 |
| GEM | 54.01 | 80.78 | 89.24 | 68.37 | 22.96 | 94.69 | 76.45 | 67.13 | 27.4 | 92.91 |
| KNN | 70.62 | 80.43 | 81.28 | 75.09 | 64.11 | 84.26 | 74.15 | 78.6 | 62.95 | 83.78 |
| CDR_KDE | 53.72 | 83.47 | 65.66 | 81.52 | 42.64 | 85.21 | 62.15 | 83.01 | 44.43 | 84.12 |
| CDR_Maha | **37.89** | **90.35** | **58.14** | **85.62** | **16.28** | **95.31** | **55.85** | **86.79** | **21.27** | **93.67** |

| Hard-OOD | Average | | ImageNet-Resize | | ImageNet-Fix | | LSUN-fix | | CIFAR10 | |
|---|---|---|---|---|---|---|---|---|---|---|
| | FPR95↓ | AUROC ↑ | FPR95↓ | AUROC↑ | FPR95↓ | AUROC↑ | FPR95↓ | AUROC↑ | FPR95↓ | AUROC↑ |
| GradNorm | 95.85 | 51.73 | 96.25 | 60.73 | 92.23 | 58.08 | 98.83 | 41.19 | 96.07 | 46.91 |
| MSP | 94.07 | 55.85 | 91.89 | 69.56 | 91.59 | 60.01 | 96.81 | 46.67 | 95.97 | **47.16** |
| Energy | 94.28 | 56.27 | 92.26 | 69.15 | 91.58 | 61.72 | 97.25 | 47.41 | 96.04 | 46.8 |
| Maha | 78.85 | 61.72 | 26.81 | 92.23 | 95.67 | 50.28 | 95.07 | 59.85 | 97.83 | 44.5 |
| GEM | 78.91 | 61.68 | 27.04 | 92.15 | 95.68 | 50.11 | 95.06 | 59.86 | 97.84 | 44.58 |
| KNN | 84.78 | **64.69** | 64.7 | 83.97 | **89.14** | **64.96** | **89.41** | **63.15** | 95.88 | 46.68 |
| CDR_KDE | 81.01 | 60.86 | 40.83 | 86.82 | 90.54 | 62.15 | 96.31 | 48.38 | 96.34 | 46.07 |
| CDR_Maha | **75.89** | 63.47 | **18.57** | **95.06** | 91.7 | 62.65 | 95.52 | 52.68 | 97.77 | 43.5 |

