# OpenReview forum: "Conditional Density Ratio Score for Post Hoc Deep Outlier Detection"
_ICLR.cc/2025/Conference — Submitted to ICLR 2025_

### Official Review · Reviewer_GZ1a · 2024-10-20

**Soundness:** 2
**Presentation:** 2
**Contribution:** 2
**Rating:** 3
**Confidence:** 5

**Summary:**

This paper introduces the Conditional Density Ratio (CDR) score, a post-hoc density estimator for out-of-distribution (OOD) detection. Unlike traditional methods that marginalize over class-conditional distributions, CDR combines a class-conditional generative model and a discriminative classifier in the latent space. A key innovation is temperature scaling, which adjusts model calibration without requiring OOD samples. CDR is compatible with estimators like the kernel density and Mahalanobis estimators. The approach is computationally efficient, requiring no additional training, and demonstrates strong performance across various OOD benchmarks. Limitations include sensitivity to poor density approximations and potential suboptimality of the temperature scaling method.

**Strengths:**

- Interesting direction for a new framework composed of previous approaches and marginal extensions.
- New way to set the temperature parameter, the optimization of which should be further pursued and evaluated
- Good results in the setup used

**Weaknesses:**

- The approach mainly combines existing work without modification. The improvements made by setting the temperature parameter are marginal improvements.
- Lack of baselines: Numerous possible baseline methods were not included in the comparison, such as [1], [2] and [3], to name just three possible ones (latest base method from 2022).
- Lack of benchmark data sets: Instead of using established benchmarks such as [4], [5] and [6], a custom benchmark scenario is used, which, in combination with the lack of baseline methods, makes comparability extremely difficult. [5] was specially designed for this purpose, as it calls this approach into question.
- In addition, [4] and [6] are cited, and [4] is even used as a baseline method, but its benchmarks are not used anyway.

**Questions:**

Q1: The introduction states that it is a general framework. The method should work with any estimator. Only the two presented are evaluated. How is this statement justified?

Q2 In Related Work, Post-Hoc Methods, the overlap with Anomaly Detection is emphasised. Why, then, is KNN the only common anomaly detection method used as a baseline?

Q3  Is the statement: "However, these generative models have generally shown poor OOD detection performance" in n Related Work, Unsupervised/Self-Supervised Methods, based only on the works cited (latest 2021)? Many more recent works contradict this.

Q4 How is the statement: "In this work, we consider a setup where no OOD samples are available for training nor inference."  to be understood? Has the method not been tested with OOD samples?

Q5 Why are so few and so old baseline methods used?

Q6 Why are none of the benchmarks mentioned used to enable better comparability?

[1] Graham, Mark S., et al. "Denoising diffusion models for out-of-distribution detection." Proceedings of the IEEE/CVF Conference on Computer Vision and Pattern Recognition. 2023.

[2] Wang, Hualiang, et al. "Clipn for zero-shot ood detection: Teaching clip to say no." Proceedings of the IEEE/CVF International Conference on Computer Vision. 2023.

[3] Jiang, Xue, et al. "Negative label guided ood detection with pretrained vision-language models." arXiv preprint arXiv:2403.20078 (2024).

[4] Rui Huang, Andrew Geng, and Yixuan Li. On the importance of gradients for detecting distributional
shifts in the wild. In NeurIPS, 2021.

[5] Guille-Escuret, Charles, et al. "Expecting The Unexpected: Towards Broad Out-Of-Distribution Detection." arXiv preprint arXiv:2308.11480 (2023).

[6] Yang, Jingkang, et al. "Openood: Benchmarking generalized out-of-distribution detection." Advances in Neural Information Processing Systems 35 (2022): 32598-32611.

---

> ### Author Response · Authors · 2024-11-20
>
> We would like to thank the reviewer for taking time to give feedback. We are encouraged by the comments that the proposed CDR method is novel, theoretically driven and versatilile (reviewer 9Qn6, 5AjN and hsYo), that the temperature scaling method is effective without OOD samples (reviewer 9Qn6 and hsYo), and that the proposed method perform well empirically (reviewer 9Qn6, hsYo, GZ1a and 5AjN). We address the comments and questions below.
>
> ### **No Novelty**
> We respectfully disagree and believe that our contribution has sufficient novelty and value. Below, we outline key points that support the originality of our work:
>    1. **Novel CDR Framework:** To the best of our knowledge, this is the first work to propose the general CDR framework for post-hoc density estimation. While the specific implementation of the CDR score leverages previously proposed estimators such as the Mahalanobis and KDE estimators, we believe this only scratches the surface of what the CDR framework can achieve. For instance, instead of using a trained classifier to derive $p(\boldsymbol{z}|y)$ through Mahalanobis or KDE estimators, an independently trained likelihood-based neural network could directly estimate $p(\boldsymbol{x}|y)$. The same CDR framework could then combine $p(\boldsymbol{x}|y)$ and $p(y|\boldsymbol{x})$ to estimate $p(\boldsymbol{x})$. We leave this as a direction for future exploration.
>    2. **Temperature Scaling:** The proposed temperature scaling method is a crucial component of the CDR framework. Figure 4 and Appendix E (Table 15) show that our temperature scaling method can significantly improve AUROC scores, increasing performance from 72% to 90% in some scenarios. Unlike prior post-hoc methods (e.g., KNN), which rely on OOD samples for hyperparameter tuning, our temperature scaling approach uses only in-distribution samples, making it more practical.
>
> ### **Lack of Baseline Methods**
> We respectfully disagree. We thank the reviewer for suggesting additional related works, which we will incorporate into our related works section [1-3]. However, we believe it may not be entirely fair to compare our framework with some of the suggested methods for the following reasons:  1. [1-2] are not post-hoc methods, as far as we know; 2. [2-3] rely on large-scale pre-trained models, whereas our method works with any classifier, including those trained from scratch. Despite these differences, we have included two recently published post-hoc OOD detection methods [5, 6] (CVPR 2023 and ICML 2024) in our evaluation. Preliminary results on CIFAR datasets are provided in the manuscript, and experiments on ImageNet are ongoing. The proposed method performs favorably against the two recently proposed method. We will update the paper with a complete set of results as soon as they are available. Lastly, we do not claim that the proposed method is state-of-the-art in post-hoc OOD detection. However, we believe the findings are worth sharing with the community for their novelty and practical value.
>
> | Inlier Data   | CIFAR10                    |                           |                           |                           | CIFAR100                  |                           |                           |                           |
> |---------------|----------------------------|---------------------------|---------------------------|---------------------------|---------------------------|---------------------------|---------------------------|---------------------------|
> | OOD Data      | Average OOD               |                           | Average Hard-OOD          |                           | Average OOD               |                           | Average Hard-OOD          |                           |
> | Metrics       | FPR95                     | AUROC                     | FPR95                     | AUROC                     | FPR95                     | AUROC                     | FPR95                     | AUROC                     |
> | GEN (Liu et al, CVPR 2023) | 37.8         | 92.45                     | 47.98                     | 89.39                     | 81.7                      | 77.60                     | 83.84                     | 74.42                     |
> | fDBD (Liu and Qin, ICML 2024) | 28.15     | 95.23                     | 45.50                     | 90.16                     | 59.05                     | 87.11                     | 77.38                     | 77.26                     |
> | CDR_KDE       | 14.68                     | 97.01                     | **38.65**                 | **91.05**                 | 30.52                     | 92.69                     | **69.87**                 | **77.91**                 |
> | CDR_Maha      | **10.53**                 | **97.72**                 | 42.36                     | 90.21                     | **19.55**                 | **96.01**                 | 71.33                     | 75.90                     |

---

> > ### Author Response · Authors · 2024-11-20
> >
> > ### **Lack of Benchmark Datasets**
> > We respectfully disagree and invite the reviewer to refer to the experiments in [4]. We used an identical experimental setup and directly obtained OOD data from the authors of [4] via their GitHub repository. To further validate the proposed method, we conducted experiments with additional model architectures (see Appendix D) and datasets, including CIFARs. While these datasets are widely used in OOD literature, we trained our own classifiers for CIFARs due to the need for an inlier validation set for temperature tuning. Since exact validation data is unavailable for pre-trained models, we believe this approach is reasonable. In addition, we also conducted additional experiments using NLP dataset. Results are summarized in Appendix C. To our knowledge, this is the first use of these NLP datasets for OOD detection evaluation. All in all, we believe we have included ample benchmark experiments to demonstrate the effectiveness and the general applicability of the proposed method.
> >
> > ### **Additional Questions**
> > Q1: The introduction states that the method is a general framework. How is this justified when only two estimators are evaluated?
> > - By "general," we mean that the proposed CDR score works with any $p(\boldsymbol{z}|y)$ and $p(y|\boldsymbol{z})$. We demonstrated this with both parametric (Gaussian mixture model) and non-parametric (KDE) estimators, which are widely representative. However, we have updated the manuscript to clarify this statement and make it less absolute.
> >
> > Q2: Why is KNN the only common anomaly detection method used as a baseline?
> > - While OOD detection and anomaly detection share similarities, they are distinct tasks. OOD detection typically involves labeled inlier samples, enabling supervised learning, whereas anomaly detection often uses unlabeled inliers and relies on unsupervised algorithms. We invite the reviewer to refer to [7] for further discussion on these differences.
> >
> > Q3: Is the statement on generative models’ poor OOD performance based on outdated citations?
> > - While deep learning is a fast moving field, we do not believe that work published in 2021 is "outdated", and still has its own merit. Nevertheless, if more recent works demonstrate that generative models outperform classifier-based OOD methods, we would be happy to include and discuss these findings in the related works section.
> >
> > Q4: The manuscript states no OOD samples are used. Has the method been tested with OOD samples?
> > - This statement refers to the temperature tuning phase, where no OOD samples are available. Many prior methods rely on OOD samples for hyperparameter tuning, but our method uses only in-distribution samples for this purpose. For evaluation and metric computation (e.g., AUROC), OOD samples are, of course, necessary. We have updated the manuscript to clarify this distinction.
> >
> > Q5: Why are so few and outdated baseline methods used?
> > - We have addressed this concern by incorporating two recent methods into our evaluations.
> >
> > Q6: Why are none of the suggested benchmarks used for comparability?
> > - As mentioned, some of the suggested methods are not post-hoc, and others rely on pre-trained models, making direct comparisons less fair. However, our method works with any classifier and demonstrates effectiveness across various datasets.
> >
> > [1] Graham, Mark S., et al. "Denoising diffusion models for out-of-distribution detection." Proceedings of the IEEE/CVF Conference on Computer Vision and Pattern Recognition. 2023.
> >
> > [2] Wang, Hualiang, et al. "Clipn for zero-shot ood detection: Teaching clip to say no." Proceedings of the IEEE/CVF International Conference on Computer Vision. 2023.
> >
> > [3] Jiang, Xue, et al. "Negative label guided ood detection with pretrained vision-language models." arXiv preprint arXiv:2403.20078 (2024).
> >
> > [4] Rui Huang, Andrew Geng, and Yixuan Li. On the importance of gradients for detecting distributional shifts in the wild. In NeurIPS, 2021.
> >
> > [5] Liu, Xixi, Yaroslava Lochman, and Christopher Zach. "Gen: Pushing the limits of softmax-based out-of-distribution detection." Proceedings of the IEEE/CVF Conference on Computer Vision and Pattern Recognition. 2023
> >
> > [6] Liu, Litian, and Yao Qin. "Fast Decision Boundary based Out-of-Distribution Detector." Forty-first International Conference on Machine Learning, 2024.
> >
> > [7] Yang, Jingkang, et al. "Generalized out-of-distribution detection: A survey." International Journal of Computer Vision (2024): 1-28.

---

> > > ### Comment · Reviewer_GZ1a · 2024-11-24
> > >
> > > Dear authors,
> > >
> > > Thank you for the response to my review. I hope to be able to clarify where possible.
> > >
> > > ---
> > >
> > > **Comment:**
> > > This statement holds true only in relation to the very limited baselines used in the paper. Such self-restriction can always lead to the desired results.
> > >
> > > **No Novelty:**
> > > This was not the point made; please read the statement carefully. However, this is only a minor reason for the given rating. Building on fundamental works is legitimate, and the underlying works are properly cited, which I appreciate. Nevertheless, the modifications remain minor.
> > >
> > > **Lack of Baselines:**
> > > The arguments seem very constructed to exclude competitive methods. I appreciate the inclusion of additional baselines and thank you for partly addressing my concerns.
> > > It is correct that the paper does not claim to achieve SOTA results, but the question remains: why should the method be used then? The argument of gaining insight feels more like “reporting negative results.” The question is whether this is sufficient as a justification in this case.
> > >
> > > **Lack of Datasets:**
> > > The use of the benchmark should at least be made explicit, and the arguments provided should be included. However, in part, the arguments again seem as though there is a deliberate focus on better-performing components. This is difficult to comprehend.
> > >
> > > **Q1:**
> > > Thank you for considering my concerns.
> > >
> > > **Q2:**
> > > What is this supposed to argue for, if kNN is included but other baselines are not? Moreover, this does not answer my question.
> > >
> > > **Q3:**
> > > “Outdated” in this context is not inherently negative; the baselines should absolutely remain in the comparison. The problem is not the older methods, but rather the absence of newer ones in such a rapidly evolving field as ML. Including baseline methods that were developed more than two years ago as the newest ones cannot be justified by any argument.
> > >
> > > **Q4:**
> > > Thank you for the clarification and correction.
> > >
> > > **Q5:**
> > > I expressly welcome this, but it only marginally mitigates the problem. To consider it fully resolved is questionable.
> > >
> > > **Q6:**
> > > As previously mentioned, these arguments for self-limitation appear very constructed to present the method’s performance in a better light.
> > >
> > > Considering the arguments, I am raising my score for the contribution. However, I cannot increase the overall score based on the responses provided. If the additional baselines are included, the good comparative performance is maintained, and more of my initial concerns are adequately addressed, I will reconsider my score for the overall evaluation.
> > >
> > > ---

---

> ### Author Response · Authors · 2024-11-25
>
> Thank you very much for your response. We sincerely appreciate your time and effort in reviewing our work. Below, we aim to further clarify our problem setup and experimental benchmarks.
>
> ### **Limited Baselines**
> We would like to reiterate that, with the inclusion of the two additional and most recent post hoc benchmark methods [2-3] according to https://github.com/Jingkang50/OpenOOD mentioned in our last response, **we believe our study encompasses a comprehensive comparison within the problem setup of “post-hoc OOD detection. with no extra data.”** See [2-5] for several recent/representative works with the post hoc problem setup. Specifically:
> 1. Problem Setup: **Our problem setup focuses on post-hoc OOD detection without the use of additional data or training. This is a common and realistic setup, widely focused on in prior works [1-5]** (Please see our paper and [1] for a comprehensive list). Within this setup, we utilize a fixed, pretrained classifier $p(y|x)$ for classification tasks such as CIFAR classification in general, with NO explicit focus on OOD detection and over which we have no control. The objective is to perform OOD detection with this fixed classifier and no additional data. We invite the reviewer to refer to Section 3.3 of the OpenOOD paper [1] for further details on this specific problem setup.
> 2. Comprehensive Benchmarks: **We have included an extensive list of benchmarks relevant to this problem setup**. Specifically:
>     * As shown in Table 1 of the OpenOOD paper [1], it is standard practice to compare methods within the same experimental setup. Our method belongs to the “OSR & OOD Detection (w/o Extra Data, w/o Training)” category, which includes numerous prior works.
>     * Consequently, we respectfully disagree with the comment that “this statement holds true only in relation to the very limited baselines used in the paper.” This problem setup is well-recognized and extensively studied.
>     * Additionally, some of our baseline benchmarks, such as the KNN methods, perform competitively even across different problem setups.
> 3. Fair Comparisons: In line with established practices in post-hoc OOD detection literature [1-5], **we compare our method against other post-hoc methods with similar experimental setups. We have not "constructed to exclude competitive methods."** Instead, we have made an effort to be as inclusive as possible. However, comparing post-hoc methods with more complex methods that require explicit training or additional datasets for OOD detection would be both unfair and impractical in many real-world scenarios. For example:
>     * Methods like [6] rely on “a vast number of negative labels from extensive corpus databases” and vision-language models, which may not be feasible in certain applications.
>     * For the above reason, methods such as [6] cannot be easily applied to CIFAR or the NLP experiments in Appendix C of our paper due to their dependence on vision-language models.
> 4. Novelty and Contribution: While we do not claim SOTA performance across the entire OOD detection literature, we believe our proposed method offers significant value:
>     * It achieves competitive performance within the well-established and widely studied post-hoc OOD problem setup we focus on.
>     * The method itself is novel and holds potential for application in other domains beyond those explored in our work. We are excited to share these findings with the broader community.

---

> > ### Author Response · Authors · 2024-11-25
> >
> > ### **Limited Datasets**
> > Thank you for the suggestion. We will revise the paper to clarify our experimental setup. However, we emphasize that **we have definitely not “deliberately focused on better-performing components” and have included a very comprehensive list of benchmark methods commonly used in OOD detection [2-5].** Most of existing post-hoc methods use similar benchmark datasets for experiments, and we have covered a pretty exhaustive list of these experiments, much more than a lot of the experiments done in some of prior literature [2-6]. On top of these common benchmarks, we have also incorporated experiments with self-supervised learning (Table 4) and NLP datasets (Appendix C) to demonstrate the generalizability of our method across different scenarios. Below are some further clarifications:
> > 1. CIFAR Experiments: The only difference between our experiments and prior work is that we trained our own classifier due to the need for a small validation set.
> > 2. ImageNet Experiments: All models were obtained directly from the TorchVision library. To further demonstrate the effectiveness of our method, for the ImageNet dataset, we tested with 7 different pretrained models (appendix D). These models were obtained directly from the TorchVision library without any fine-tuning or hyperparameter adjustments, and similar conclusions generally hold. To the best of our knowledge, this is a very comprehensive study, and most prior works focused on much fewer number of models for experiments.
> > 3. NLP Experiments (Appendix C): Embeddings were directly obtained from the ADBench dataset [7].
> > 4. No Hyperparameter Tuning: We did not tune hyperparameters; the experimental setup remained consistent across all experiments.
> >
> > ### **additional questions**
> > Q2:
> > We would really appreciate if the reviewer can be more specific in what your question means so that we can better address the issue. In general, it is impossible to compare with all the benchmark methods across all problem setups in the world, and argued above, we believe we have already included a very comprehensive list of experiments.
> >
> > Q3:
> > Definitely. We will update the related works accordingly. We would really appreciate if the reviewer can point us towards some of the references which compare generative methods with classifier methods if you have any specific work in mind.
> >
> > Q5 - Q6:
> > Please see above.
> >
> > #### **References**
> >
> > [1] Yang, Jingkang, et al. "OpenOOD: Benchmarking generalized out-of-distribution detection." Advances in Neural Information Processing Systems, 35 (2022): 32598-32611.
> >
> > [2] Liu, Litian, and Yao Qin. "Fast Decision Boundary based Out-of-Distribution Detector." International Conference on Machine Learning, 2024.
> >
> > [3] Liu, Xixi, Yaroslava Lochman, and Christopher Zach. "Gen: Pushing the limits of softmax-based out-of-distribution detection." Proceedings of the IEEE/CVF Conference on Computer Vision and Pattern Recognition, 2023.
> >
> > [4] Sun, Yiyou, et al. "Out-of-distribution detection with deep nearest neighbors." International Conference on Machine Learning. PMLR, 2022.
> >
> > [5] Liu, Weitang, et al. "Energy-based out-of-distribution detection." Advances in Neural Information Processing Systems, 33 (2020): 21464-21475.
> >
> > [6] Jiang, Xue, et al. "Negative label guided OOD detection with pretrained vision-language models." arXiv preprint arXiv:2403.20078 (2024).
> >
> > [7] Han, Songqiao, et al. "ADBench: Anomaly Detection Benchmark." Advances in Neural Information Processing Systems, 35 (2022): 32142-32159.

---

> ### Comment · Reviewer_GZ1a · 2024-11-26
> **Reply**
>
> Thanks to the Authors for the quick response.
>
> ---
>
> **Limited Baselines**
> 1.
> This is less about the setup itself and more about the questionable nature of such self-imposed limitations to post-hoc methods. A method should be compared under identical conditions (dataset and split) against all methods that can operate under the same conditions. I do not expect comparisons to methods under entirely different settings, but many OOD and anomaly detection (e.g. from ADBench) methods, including those mentioned in the review, could be trained and theoretically and practically compared under the same conditions. Not all of them, but at least the top-performing methods.
>
> 2.
> The construction of the benchmark should at least be more clearly justified and described in the paper. At a minimum, the explanations provided here should be made more explicit in the paper.
>
> 3.
> See Q2.
>
> 4.
> I appreciate the improvements in the new version very much. However, once again, this is not the primary reason for the score.
>
> To clarify again: the issue is not with the experiments that were conducted or the baselines and literature used, but rather with the absence of the extensive, more recent literature and methods (see [1] and references from the review).
>
> ---
>
> **Lack of Datasets**
> Once again, I am not criticizing the experiments conducted. The issue is what has been omitted. In my view, there are two ways to establish comparability with the literature: either by using the same setup and datasets or by including the methods from these works. Neither was done here to a sufficient extent.
>
> ---
>
> **Q2**
> The original question remains: *“In Related Work, Post-Hoc Methods, the overlap with Anomaly Detection is emphasized. Why, then, is KNN the only common anomaly detection method used as a baseline?”*
> To clarify further: You attempt to justify the exclusion of additional baselines by referencing post-hoc methods. However, KNN is not a post-hoc method, so it seems acceptable to include non-post-hoc methods as baselines. Now, while KNN is not a poor anomaly detection method, it quickly reaches its limits, particularly in high-dimensional problems like this one. There are countless other (and superior) anomaly detection methods that could serve as baselines and would perform better in high-dimensional problems as here. Hence, once more: why was KNN chosen as a baseline method while other, more competitive anomaly detection methods were not?
>
> ---
>
> **Q3**
> I recommend this work: [1], [4]. Also [6] is possible to consider (ICLR 24). Even if the focus here is not on new methods for OOD detection, the ideas should be taken into account. Futhermore, this work could be used for identification [2]. Even with your given limits, there are other baselines such as [3] and the baselines used in their paper (the non-overlapping ones). And for different methods [5].(other of many much newer works) I also recommend this git https://github.com/huytransformer/Awesome-Out-Of-Distribution-Detection?tab=readme-ov-file#ood-detection for countless other papers from top conferences, many of them from the years after 2022.
>
> ---
>
> **Reminder**
> As I mentioned, despite my remaining concerns and the observed limitations, I will reconsider my score once the new version, including additional baseline methods, is published. Although there are still justified doubts and lingering concerns about completeness, I appreciate the additional experiments and inclusion of further baselines.
>
> ---
>
> [1] Sharifi, Sina et al. “Gradient-Regularized Out-of-Distribution Detection.” European Conference on Computer Vision (2024).
>
> [2] Yang, Jingkang, et al. "Generalized out-of-distribution detection: A survey." International Journal of Computer Vision (2024): 1-28.
>
> [3] Tang, Keke, et al. "CORES: Convolutional Response-based Score for Out-of-distribution Detection." Proceedings of the IEEE/CVF Conference on Computer Vision and Pattern Recognition. 2024.
>
> [4] Chen, Jiankang, et al. "FodFoM: Fake Outlier Data by Foundation Models Creates Stronger Visual Out-of-Distribution Detector." Proceedings of the 32nd ACM International Conference on Multimedia. 2024.
>
> [5] Le Bellier, Georges, and Nicolas Audebert. "Detecting Out-Of-Distribution Earth Observation Images with Diffusion Models." Proceedings of the IEEE/CVF Conference on Computer Vision and Pattern Recognition. 2024.
>
> [6] Jaini, Priyank, Kevin Clark, and Robert Geirhos. "Intriguing properties of generative classifiers." arXiv preprint arXiv:2309.16779 (2023).

---

> > ### Author Response · Authors · 2024-11-27
> >
> > We thank the reviewer for their reply and would appreciate clarification on your comments provided. Below, we address each point raised.
> >
> > ***Limited Baselines***
> >
> > “Absence of the extensive, more recent literature and methods.”
> > With due respect, could the reviewer clarify why the additional experiments conducted using the two recent methods [1, 2] (published in CVPR 2023 and ICML 2024) are not considered “recent”? For your reference:
> > 1. Both works are included in the list of https://github.com/huytransformer/Awesome-Out-Of-Distribution-Detection?tab=readme-ov-file#ood-detection.
> > 2. We selected these methods because they align with the same problem setup as our proposed method and are also listed in the https://github.com/Jingkang50/OpenOOD.
> > We have not “handpicked” methods that happen to perform worse than ours. These two were selected because they have the same experimental setup as ours.
> > 3. It would be helpful if the reviewer could clarify how “recent” is defined and why works from CVPR 2023 and ICML 2024 are not valid for comparison.
> >
> > In addition, we would like to note that the works cited by the reviewer are not the most suitable benchmark methods for comparison. Specifically,
> > - [5]: This method is not practical in our setup because it assumes access to an auxiliary OOD dataset, which we do not have.
> >
> > - [3]: This method is not the best benchmark for comparison because it assumes the model must be a convolutional network. In contrast, our proposed framework is a general method that can be applied to any neural network classifier, making it more versatile.
> >
> > - [6]: This method is impractical for our setup as it assumes access to “multiple foundation models to generate two types of challenging fake outlier images for classifier training.” Such a requirement introduces a very stringent assumption that is not feasible in many real-world scenarios.
> >
> > In general, we do believe that, we need to use identical setup across all methods for a fair comparison, and it is unfair to compare methods with additional requirements. We also would like to emphasize that the problem setup studied in this paper is a widely acknowledged one, as argued above.
> >
> > ***Lack of Dataset***
> >
> > “There are two ways to establish comparability with the literature: either by using the same setup and datasets or by including the methods from these works. Neither was done here to a sufficient extent.”
> > We respectfully disagree with this comment. **We are using an identical setup (same dataset, same model, varying only the method for OOD detection) for all methods tested.** Additionally, we have included a broad set of post-hoc methods as benchmarks.
> > Could the reviewer clarify explicitly what aspect of our comparison is not “identical”? Our intention was to adhere closely to standard practices in post-hoc OOD detection evaluation, and any specific and concrete feedback on this point would be appreciated. We are more than happy to update the paper to clarify any confusion.
> >
> > ***Regarding KNN***
> >
> > The reason we used KNN is precisely because it has been proposed as a post-hoc method for OOD detection [4]. All of our benchmark methods are post-hoc methods so that we can compare fairly with identical experimental setup. The comment that “it quickly reaches its limits, particularly in high-dimensional problems like this one” is not directly applicable here. Empirically, KNN has demonstrated strong performance across a variety of benchmark datasets. Furthermore, we applied KNN in the lower-dimensional feature representation space (i.e., the penultimate layer of the classifier network), where the dimensionality issue is significantly mitigated.

---

> ### Author Response · Authors · 2024-11-27
>
> ***Reminder***
>
> We would be happy to include additional experiments from other recent works if the reviewer can justify why [1, 2] are insufficient. However, it may not be feasible to update the paper at this stage, as the deadline to upload a revised PDF is tomorrow. That said, we are happy to provide additional details in the comments for now and incorporate any necessary updates in future versions of the paper if applicable.
>
> ***Paper Revision***
>
> We have updated the manuscript to include the clarifications on our experiments. Please see section 4 for details. We would be happy to make further adjustments to the paper if there are any additional concerns and if time permits.
>
> Lastly, we emphasize that, we are very confident that we have conducted a fair comparison of our method against the most suitable benchmark methods using identical experimental setups for all methods implemented.
>
> [1] Liu, Litian, and Yao Qin. "Fast Decision Boundary based Out-of-Distribution Detector." International Conference on Machine Learning, 2024.
>
> [2] Liu, Xixi, Yaroslava Lochman, and Christopher Zach. "Gen: Pushing the limits of softmax-based out-of-distribution detection." Proceedings of the IEEE/CVF Conference on Computer Vision and Pattern Recognition, 2023.
>
> [3] Tang, Keke, et al. "CORES: Convolutional Response-based Score for Out-of-distribution Detection." Proceedings of the IEEE/CVF Conference on Computer Vision and Pattern Recognition. 2024.
>
> [4] Sun, Yiyou, et al. "Out-of-distribution detection with deep nearest neighbors." International Conference on Machine Learning. PMLR, 2022.
>
> [5] Sharifi, Sina et al. “Gradient-Regularized Out-of-Distribution Detection.” European Conference on Computer Vision (2024).
>
> [6] Chen, Jiankang, et al. "FodFoM: Fake Outlier Data by Foundation Models Creates Stronger Visual Out-of-Distribution Detector." Proceedings of the 32nd ACM International Conference on Multimedia. 2024.

---

> > ### Comment · Reviewer_GZ1a · 2024-11-28
> > **Reply**
> >
> > Thank you for the timely response.
> >
> > I have presented numerous options, and I think it is clear that you will always find some reason to exclude additional baselines that would ensure true comparability. Once again, any method capable of operating under the same conditions should be included. Restricting the scope so narrowly that only methods from the same field with precisely the same mode of operation are included, while excluding all others capable of working under the same conditions, makes no sense.
> >
> > Furthermore, I was appalled to discover that the additional baselines results mentioned in the authors' initial response during this discussion were not included in the paper. As a result, the response is rendered meaningless, and the experiments must be considered as not having been conducted.
> >
> > The basis for evaluation is the content of the paper, not the implications of results suggested during the discussion. Consequently, despite the general improvements made to the paper, I have decided not to further increase my rating.

---

> > > ### Author Response · Authors · 2024-11-28
> > > **Concluding Remarks to Reviewer GZ1a**
> > >
> > > As stated since the beginning, **we have been open to including additional benchmarks that ensure a fair comparison, hence the two additional recent baselines included above. The reviewer never justified why these benchmarks are "invalid"**. "It is clear that the reviewer will always find some reason to disregard" the numerous benchmarks we have included.
> > >
> > > **Performing experiments and comparisons with methods under the identical setup precisely ensures apples-to-apples comparison and "true comparability", and allows us to understand transparently that improvements are coming from methodology** (and not from additional data or foundation models, which are used in methods the reviewer has asked for). We believe this is good practice in research and hope our methodology will inspire future work.
> > >
> > > **It is unfortunate that the reviewer chose to regard our response here as "meaningless"**. This is a public review forum and we take the results and discussions presented here seriously. As we have openly stated above, the additional benchmark results will be added to the final paper once all experiments have concluded.
> > >
> > > Finally, to reiterate the important line of research on OOD detection without additional training or auxiliary dataset, we provide again here a list of representative recent works [2-10], a lot of which serving as our benchmark methods in this paper. We believe, the fact that our problem setup is considered as a standalone group in the OpenOOD benchmark [1] (See Table 1: OSR & OOD Detection (w/o Extra Data, w/o Training) suggests the validity of our problem setup. More works can be found in Section 3.1.1.a of [12].
> > >
> > > [1] Yang, Jingkang, et al. "Openood: Benchmarking generalized out-of-distribution detection." Advances in Neural Information Processing Systems 35 (2022): 32598-32611
> > >
> > > [2] Liu, Litian, and Yao Qin. "Fast Decision Boundary based Out-of-Distribution Detector." International Conference on Machine Learning, 2024.
> > >
> > > [3] Liu, Xixi, Yaroslava Lochman, and Christopher Zach. "Gen: Pushing the limits of softmax-based out-of-distribution detection." Proceedings of the IEEE/CVF Conference on Computer Vision and Pattern Recognition, 2023.
> > >
> > > [4] Sun, Yiyou, et al. "Out-of-distribution detection with deep nearest neighbors." International Conference on Machine Learning. PMLR, 2022.
> > >
> > > [5] Hendrycks, Dan, and Kevin Gimpel. "A baseline for detecting misclassified and out-of-distribution examples in neural networks." arXiv preprint arXiv:1610.02136 (2016).
> > >
> > > [6] Djurisic, Andrija, et al. "Extremely simple activation shaping for out-of-distribution detection." arXiv preprint arXiv:2209.09858 (2022).
> > >
> > > [7] Liu, Weitang, et al. "Energy-based out-of-distribution detection." Advances in neural information processing systems 33 (2020): 21464-21475.
> > >
> > > [8] Morteza, Peyman, and Yixuan Li. "Provable guarantees for understanding out-of-distribution detection." Proceedings of the AAAI Conference on Artificial Intelligence. Vol. 36. No. 7. 2022.
> > >
> > > [9] Sun, Yiyou, and Yixuan Li. "Dice: Leveraging sparsification for out-of-distribution detection." European Conference on Computer Vision. Cham: Springer Nature Switzerland, 2022.
> > >
> > > [10] Wang, Haoqi, et al. "Vim: Out-of-distribution with virtual-logit matching." Proceedings of the IEEE/CVF conference on computer vision and pattern recognition. 2022.
> > >
> > > [11] Huang, Rui, Andrew Geng, and Yixuan Li. "On the importance of gradients for detecting distributional shifts in the wild." Advances in Neural Information Processing Systems 34 (2021): 677-689.
> > >
> > > [12] Yang, Jingkang, et al. "Generalized out-of-distribution detection: A survey." International Journal of Computer Vision (2024): 1-28.

---

> ### Comment · Reviewer_GZ1a · 2024-11-29
> **Reply**
>
> You have had ample time throughout this discussion to incorporate, for instance, the additional experiments mentioned in your initial response into the paper itself. Furthermore, your cited sources confirm the possibility of including numerous other baselines. If, as you claim, these comparisons are valid, then the newer methods presented in these references could also have been included as baselines.
>
> It would have been sufficient if you had incorporated either the two methods outlined in your initial response or any of those mentioned in your references into the results section of the paper as baselines. However, you did not do this. Consequently, none of this is reflected in the paper and cannot, therefore, be considered as part of the evaluation criteria. The baselines did not need to be the ones I originally proposed—just additional and particularly more recent ones (more recent than the newest method you used from 2022, such as those cited in your references (e.g. Liu, Litian, and Yao Qin. "Fast Decision Boundary based Out-of-Distribution Detector." International Conference on Machine Learning, 2024.). This work that you cited is not limited to the scope post-hoc methods you mentioned and compares against different methods, they also do not frame KNN as post-hoc method (see Section 4. Experiments, Baselines, for KNN you can also see Yang, Jingkang, et al. "Generalized out-of-distribution detection: A survey." International Journal of Computer Vision (2024): 1-28). For references from 2022 or earlier, it goes without saying that they cannot use more recent work as a baseline if you refer to it.
>
> The basis for my recommendation must be the final version of the paper. As it stands, none of these additional methods were included in the results section, apart from a few minor textual adjustments. To reiterate clearly: there were no additions to the results section compared to the original version. This, along with the evidence provided in your references, reinforces my assessment, which must be based primarily on the content of the paper itself, not on the discussion.
>
> The discussion cannot address the persistent shortcomings of the paper if the additional results and methods cited here as baselines are not reflected in the paper's practical results (beyond adding references). Apart from a few small textual changes, none of my original concerns have been addressed. Therefore, it is not possible to raise the overall evaluation score.

---

> > ### Author Response · Authors · 2024-11-30
> > **Clarifying Misunderstandings**
> >
> > Thank you very much for your detailed replies. We appreciate your time spent reviewing our paper. Upon reviewing your latest comments and revisiting earlier discussions, we believe there has been a misunderstanding. We hope to clarify and apologize for any potential miscommunications regarding the term “additional baselines” in our prior exchanges.
> >
> > **“It would have been sufficient if you had incorporated either of the two methods outlined in your initial response.”**
> > First, we thank the reviewer for recognizing the validity of our experimental setup and the additional benchmark methods we chose. However, it appears there was a misunderstanding on our part regarding the reviewer’s earlier comments, which stated: *“If the additional baselines are included, I will reconsider my score for the overall evaluation”* and *“I will reconsider my score once the new version, including additional baseline methods, is published.”*
> >
> > We mistakenly interpreted “additional baseline methods” as referring to the methods suggested by the reviewer beyond the two additional baseline methods we already included during the discussion period. As a result, we focused primarily on addressing the comparability of the suggested baselines and responding to concerns rather than completing experiments with the two additional methods already included.
> >
> > For the same reason, we misunderstood the reviewer’s reference to “publishing” to mean posting results from methods beyond those we have provided into the forum, rather than finalizing results from the two existing additional baselines provided in the forum and adding them to the paper. It has been our impression that the discussions serve as an integral part of the reviewing process and that there would be time to amend the paper for the camera-ready version if applicable.
> >
> > Once again, we sincerely apologize for this confusion and would like to reiterate that the results provided above will be updated in the revised paper. Due to this misunderstanding, we missed the opportunity to include these updates in the latest submission, and we ask for the reviewer’s understanding. We will also complete any pending experiments with the two additional baselines and provide updates during the discussion period, time permitting. Any final results will be incorporated into the camera-ready version if applicable.
> >
> > **Minor Clarifications Regarding KNNs:**
> > We also thank the reviewer for carefully reading the references provided. To the best of our understanding, the authors of fDBD (Liu, Litian, and Yao Qin. *"Fast Decision Boundary Based Out-of-Distribution Detector."* International Conference on Machine Learning, 2024) distinguish their method from KNNs because KNN requires an OOD dataset to tune the hyper-parameter K. Both methods, including ours and fDBD, are post-hoc in nature, utilizing a pre-trained classifier without fine-tuning. However, compared to KNN, methods like ours and fDBD offer significant advantages. For instance, fDBD is hyper-parameter-free, and our method allows for automatic tuning using the original inlier-only training/validation dataset (i.e., for the temperature term in our proposed method via the proposed tuning strategy).
> >
> > When comparing with KNN, we used the default (tuned) K proposed in the original paper since we do not assume access to OOD data for hyper-parameter tuning. We believe this distinction highlights the strengths of our approach and explains our focus on methods with a similar problem setup.

---

> > > ### Author Response · Authors · 2024-12-01
> > > **Updates on Additional Experiments.**
> > >
> > > We include here the comprehensive results using CIFAR10, CIFAR100 and Imagenet as inliers with the two additional benchmarks GEN (CVPR 2023) and fDBD (ICML 2024), and their comparisons against our proposed methods. We use identical experimental setup for all experiments conducted. The results correspond to Table 1, 2 and 3 of the paper respectively. We promise to include the results shown below as it is to the camera-ready version of the paper, if applicable. In general, our methods perform competitively against the two additional recent benchmarks. We hope that the results below can make the reviewer feel more confident about our proposed method.
> > >
> > > ### ***CIFAR10*** (Table 1)
> > > | OOD       | Average   |          | SVHN       |          | LSUN-Resize |          | Texture    |          | iSUN       |          |
> > > |-----------|-----------|----------|------------|----------|-------------|----------|------------|----------|------------|----------|
> > > |           | FPR95 ↓   | AUROC ↑  | FPR95 ↓    | AUROC ↑  | FPR95 ↓     | AUROC ↑  | FPR95 ↓    | AUROC ↑  | FPR95 ↓    | AUROC ↑  |
> > > | GEN       | 37.78     | 92.46    | 46.80      | 91.37    | 24.37       | 95.78    | 53.94      | 87.34    | 26.00      | 95.34    |
> > > | fDBD      | 28.15     | 95.23    | 35.61      | 94.82    | 19.78       | 96.55    | 35.18      | 93.39    | 22.01      | 96.17    |
> > > | CDR_KDE   | 14.68     | 97.01    | 19.71      | 96.51    | 7.17        | 98.59    | 23.94      | 94.52    | 7.90       | 98.43    |
> > > | CDR_Maha  | **10.53** | **97.72**| **11.87**  | **97.75**| **5.37**    | **98.80**| **18.42**  | **95.70**| **6.47**   | **98.64**|
> > > ---
> > > | Hard-OOD  | Average   |          | ImageNet-Resize |          | ImageNet-Fix |          | LSUN-fix  |          | CIFAR100   |          |
> > > |-----------|-----------|----------|-----------------|----------|--------------|----------|-----------|----------|------------|----------|
> > > |           | FPR95 ↓   | AUROC ↑  | FPR95 ↓         | AUROC ↑  | FPR95 ↓      | AUROC ↑  | FPR95 ↓   | AUROC ↑  | FPR95 ↓    | AUROC ↑  |
> > > | GEN       | 47.99     | 89.39    | 35.93           | 93.17    | 50.04        | 88.80    | 49.78     | 89.48    | 56.19      | 86.12    |
> > > | fDBD      | 45.50     | 90.16    | 29.02           | 94.61    | 47.02        | 90.08    | 51.69     | 88.63    | 54.27      | **87.31**    |
> > > | CDR_KDE   | **38.65** | **91.05**| 13.33           | 97.26    | **43.85**    | **90.13**    | **44.44** | **90.62**| **52.97**  | 86.19    |
> > > | CDR_Maha  | 42.36     | 90.21    | **10.23**       | **97.91**| 49.82        | 89.15    | 50.72     | 89.53    | 58.67      | 84.24    |
> > >
> > > ### ***CIFAR100*** (Table 2)
> > > | OOD       | Average   |          | SVHN       |          | LSUN-Resize |          | Texture    |          | iSUN       |          |
> > > |-----------|-----------|----------|------------|----------|-------------|----------|------------|----------|------------|----------|
> > > |           | FPR95 ↓   | AUROC ↑  | FPR95 ↓    | AUROC ↑  | FPR95 ↓     | AUROC ↑  | FPR95 ↓    | AUROC ↑  | FPR95 ↓    | AUROC ↑  |
> > > | GEN       | 81.70     | 77.60    | 83.24      | 78.06    | 76.99       | 80.81    | 87.23      | 72.71    | 79.33      | 78.81    |
> > > | fDBD      | 59.05     | 87.11    | 73.23      | 83.28    | 47.09       | 90.35    | 64.01      | 85.82    | 51.88      | 88.98    |
> > > | CDR_KDE   | 30.52     | 92.69    | 52.29      | 87.72    | 18.68       | 95.47    | 29.65      | 92.83    | 21.47      | 94.73    |
> > > | CDR_Maha  | **19.55** | **96.01**| **37.88**  | **93.19**| **9.27**    | **97.84**| **19.27**  | **95.81**| **11.79**  | **97.19**|
> > > ---
> > > | Hard-OOD  | Average   |          | ImageNet-Resize |          | ImageNet-Fix |          | LSUN-fix  |          | CIFAR10    |          |
> > > |-----------|-----------|----------|-----------------|----------|--------------|----------|-----------|----------|------------|----------|
> > > |           | FPR95 ↓   | AUROC ↑  | FPR95 ↓         | AUROC ↑  | FPR95 ↓      | AUROC ↑  | FPR95 ↓   | AUROC ↑  | FPR95 ↓    | AUROC ↑  |
> > > | GEN       | 83.84     | 74.43    | 81.65           | 76.02    | 79.45        | 76.89    | 90.87     | 69.57    | **83.37**      | **75.23**|
> > > | fDBD      | 77.38     | 77.26    | 54.86           | 88.20    | 78.82        | 77.16    | 90.92     | 69.38    | 84.92      | 74.28    |
> > > | CDR_KDE   | **69.87** | **77.91**| 23.85           | 94.06    | **77.22**    | **77.29**| 90.77     | 67.98    | 87.65      | 72.30    |
> > > | CDR_Maha  | 71.33     | 75.90    | **12.34**       | **97.26**| 84.51        | 74.15    | **93.06**     | **65.66**    | 95.42      | 66.52    |

---

> > > > ### Author Response · Authors · 2024-12-01
> > > > **Updates on Additional Experiments.**
> > > >
> > > > ### ***ImageNet*** (Table 3)
> > > > | Method    | Average    |          | Places     |          | SUN        |          | iNaturalist |          | Texture    |          |
> > > > |-----------|------------|----------|------------|----------|------------|----------|-------------|----------|------------|----------|
> > > > |           | FPR95 ↓    | AUROC ↑  | FPR95 ↓    | AUROC ↑  | FPR95 ↓    | AUROC ↑  | FPR95 ↓     | AUROC ↑  | FPR95 ↓    | AUROC ↑  |
> > > > | GEN       | 55.68      | 87.63    | 63.34      | 85.30    | **59.36**  | 87.01    | **40.43**   | 93.24    | 59.57      | 84.98    |
> > > > | fDBD      | 51.78      | 89.83    | **62.88**  | **86.20**| 59.70      | 88.24    | 40.65       | **93.79**| 43.87      | 91.09    |
> > > > | CDR_KDE   | **45.61**  | **91.02**| 63.22      | 86.07    | **59.36**  | **88.39**| 43.74       | 93.47    | **16.13**      | **96.15**|
> > > > | CDR_Maha  | 64.82      | 85.61    | 83.66      | 78.62    | 81.90      | 81.08    | 71.16       | 86.74    | 22.55      | 96.00    |

---

> > > > > ### Comment · Reviewer_GZ1a · 2024-12-03
> > > > > **Reply**
> > > > >
> > > > > I'm sorry for the misunderstanding regarding the additional experiments. However, as you have noticed, I have pointed this out several times. Since the paper has to be the basis for the evaluation, I can't increase the score, but I would like to expressly praise the additional experiments.

---

> > > > > > ### Author Response · Authors · 2024-12-03
> > > > > >
> > > > > > We thank the reviewer for the continuous engagement in the discussion. We are glad the reviewer is satisfied with the additional experiments. We have included the results above in our paper manuscript and will publish it to the camera-ready version of the paper if applicable. As such, we sincerely hope the reviewer can reconsider the decision to maintain the final score.

---

### Official Review · Reviewer_5AjN · 2024-10-31

**Soundness:** 3
**Presentation:** 2
**Contribution:** 2
**Rating:** 6
**Confidence:** 3

**Summary:**

This paper proposed a principled post hoc density estimator called Conditional Density Ratio (CDR) score, which is composed of a class-conditional generative model and a discriminative classifier model. The proposed method can estimate the marginal densities of the latent representation without marginalization. Besides, the authors proposed to automatically tune the temperature parameter to balance the relative contributions of the generative and discriminative models. Sufficient experiments were conducted to prove the effectiveness of the proposed CDR score and temperature adjustment method.

**Strengths:**

1. The paper gave a detailed theoretical introduction to the proposed CDR score and temperature tuning method.
2. The author conducted sufficient comparative experiments with benchmark methods to prove the effectiveness of the proposed CDR score, and did ablation experiments to demonstrate the importance of temperature tuning.

**Weaknesses:**

1. The motivation of the paper is unclear. The author did not explain clearly the advantages of the proposed CDR Score compared with existing post hoc density estimators.
2. The related work is not focused on "Post-Hoc Methods".
3. Some of the content in the method section and the experiment section is difficult to follow.
4. There are several errors in the paper. For instance, in Section 5, the statement "we show plots of AUROC vs. temperature for various datasets in Figure 3" should be corrected to "Figure 4".

**Questions:**

Here are some further details about weakness:
1. The motivation of the paper is unclear.
In the abstract, the authors state: "the existing post hoc density estimation has mainly focused on marginalizing the conditional distributions over all possible classes." However, the CDR Score proposed in this paper enables us to "estimate the marginal densities of the latent representation without marginalization." It seems that the advantage of CDR is that it does not require marginalization. But what advantages can avoiding marginalization bring? Will it reduce costs? Or will it make the method more suitable for application scenarios? The authors should give more specific advantages.
2. The related work is not focused on "Post-Hoc Methods".
As the method proposed in this paper is a kind of post-hoc method, the related work section should focus on explaining the shortcomings of the existing post-hoc methods and the improvements that the method proposed in this paper can bring.
3. Some of the content in the method section and the experiment section is difficult to follow
(1) The authors stated that "In this work, we propose a unified approach that combines MSP and Mahalanobis distance in a principled way for OOD detection." However, in Section 3, the authors did not explain how the CDR Score combines MSP and Mahalanobis distance.
(2) Figure 2 is difficult to follow. What is the meaning of "in" and "out"?

---

> ### Author Response · Authors · 2024-11-20
>
> We would like to thank the reviewers for taking the time to read our paper and provide valuable feedback. We are encouraged by the comments recognizing that the proposed CDR method is novel, theoretically driven, and versatile (reviewers 9Qn6, 5AjN, and hsYo); that the temperature scaling method is effective without OOD samples (reviewers 9Qn6 and hsYo); and that the proposed method performs well empirically (reviewers 9Qn6, hsYo, GZ1a, and 5AjN). Below, we address the comments, suggestions, and questions in detail.
>
> ### **Unclear Motivation: Advantage of CDR over Existing Post-Hoc Density Estimators**
> We thank the reviewer for this excellent suggestion. We acknowledge that the advantages of CDR over existing post-hoc density estimators, particularly marginalization-based methods, were not explicitly stated. Below, we summarize the key advantages of the proposed method:
>
>    1. **Novelty:** As an exploratory work, we believe the proposed CDR score itself is a significant contribution. Most existing methods rely on marginalization to estimate the marginal density of a distribution, while the CDR score explores an underexplored direction by avoiding marginalization entirely.
>
>    2. **Principled Combination of Two Approaches:** The CDR score offers a principled method to combine $p(y|\boldsymbol{z})$ and  $p(\boldsymbol{z}|y)$, both of which have been independently used for OOD detection. By combining these two approaches, the CDR score achieves better performance than either method used in isolation. Furthermore, beyond the post-hoc OOD detection setup, we believe the CDR score can serve as a general framework for density estimation. For instance, instead of using a trained classifier to estimate $p(\boldsymbol{z}|y)$, an independently trained likelihood-based neural network could estimate $p(\boldsymbol{x}|y)$. The CDR framework could then combine $p(\boldsymbol{x}|y)$ and $p(y|\boldsymbol{x})$ to estimate $p(\boldsymbol{x})$ more effectively. We leave this promising direction for future exploration.
>
>    3. **Robustness to Partition Function Issues:** From a theoretical perspective, the energy score is only a valid density estimator if the partition function for normalization is constant. This assumption often fails in practice, leading to inaccuracies. In contrast, the CDR score does not rely on marginalization, allowing it to function as a density estimator even when the partition function is not constant.
>
>    4. **Empirical Effectiveness:** The proposed method demonstrates strong empirical performance compared to existing marginalization-based methods, as shown in our experiments.
>
> We have added a dedicated subsection in the methods section to clearly discuss these advantages.
>
> ### **Related Work: Need for More Comparison with Existing Post-Hoc Methods**
> We have revised the related works section to include additional discussions comparing the CDR score with existing post-hoc methods, particularly marginalization-based density estimation methods.
>
> ### **Difficulty in Following the Method and Experiment Sections**
> We have revised both the method and experiment sections to improve readability. To further facilitate understanding, we provide a brief summary of our contributions here. We propose the CDR score, a novel approach to estimating density scores for OOD detection using a trained classifier. This approach relies on $p(y|\boldsymbol{z})$, directly obtained from the classifier, and $p(\boldsymbol{z}|y)$, which requires additional estimation. To estimate $p(\boldsymbol{z}|y)$, we explore the use of both KDE and Mahalanobis estimators. These estimations are computationally efficient, requiring only a trained feature extractor and a small validation set. We conduct extensive experiments on commonly used benchmark datasets to demonstrate the effectiveness of the proposed method.

---

> ### Author Response · Authors · 2024-11-20
>
> ### **Additional Post-Hoc OOD Detection Methods**
> We have implemented two recently proposed post-hoc OOD detection methods for comparison. Preliminary results on CIFAR datasets are provided in the table below. In general, the proposed CDR score performs competitively against these methods. We are still finalizing the experiments and will update the paper with the complete results once they are available.
>
> | Inlier Data   | CIFAR10                    |                           |                           |                           | CIFAR100                  |                           |                           |                           |
> |---------------|----------------------------|---------------------------|---------------------------|---------------------------|---------------------------|---------------------------|---------------------------|---------------------------|
> | OOD Data      | Average OOD               |                           | Average Hard-OOD          |                           | Average OOD               |                           | Average Hard-OOD          |                           |
> | Metrics       | FPR95                     | AUROC                     | FPR95                     | AUROC                     | FPR95                     | AUROC                     | FPR95                     | AUROC                     |
> | GEN (Liu et al, CVPR 2023) | 37.8         | 92.45                     | 47.98                     | 89.39                     | 81.7                      | 77.60                     | 83.84                     | 74.42                     |
> | fDBD (Liu and Qin, ICML 2024) | 28.15     | 95.23                     | 45.50                     | 90.16                     | 59.05                     | 87.11                     | 77.38                     | 77.26                     |
> | CDR_KDE       | 14.68                     | 97.01                     | **38.65**                 | **91.05**                 | 30.52                     | 92.69                     | **69.87**                 | **77.91**                 |
> | CDR_Maha      | **10.53**                 | **97.72**                 | 42.36                     | 90.21                     | **19.55**                 | **96.01**                 | 71.33                     | 75.90                     |
>
> Lastly, we have updated the manuscript to address all the issues pointed out in the review. We invite the reviewer to examine the revised version and provide any additional feedback. We sincerely appreciate the thoughtful and constructive comments, which have greatly improved the quality of our paper.
>
> [1] Liu, Litian, and Yao Qin. "Fast Decision Boundary based Out-of-Distribution Detector." Forty-first International Conference on Machine Learning, 2024.
>
> [2] Liu, Xixi, Yaroslava Lochman, and Christopher Zach. "Gen: Pushing the limits of softmax-based out-of-distribution detection." Proceedings of the IEEE/CVF Conference on Computer Vision and Pattern Recognition. 2023.

---

> > ### Comment · Reviewer_5AjN · 2024-11-25
> >
> > The authors have solved most of my doubts, and the paper has been improved. The motivation of the revised paper is clearer. Specifically, the author analyzes the advantages of the CDR method proposed in this paper compared with the marginalization-based method from a theoretical perspective (i.e., robustness to partition function issues). In addition, in the related work section, the authors highlighted the advantages of CDR method compared with existing post-hoc methods.
> >
> > Considering that the paper has been improved, the rating would be increased to 6.

---

> > > ### Author Response · Authors · 2024-11-28
> > >
> > > Thank you again for your helpful comments. We appreciate it very much.

---

### Official Review · Reviewer_hsYo · 2024-11-01

**Soundness:** 3
**Presentation:** 3
**Contribution:** 3
**Rating:** 8
**Confidence:** 3

**Summary:**

This paper proposes the Conditional Density Ratio (CDR) score, a new approach for post-hoc OOD detection in deep learning models. The key scenario addressed is when only pre-trained model weights and a small number of in-distribution validation samples are available, without access to OOD samples or training data.
The authors propose new principled framework for post-hoc OOD detection called the CDR score, which combines a class-conditional generative model in the latent space, a discriminative classifier model and a method to estimate marginal densities without class marginalization. The paper also introduce a novel temperature scaling technique that automatically calibrates the relative contributions of the generative and discriminative models by using in-distribution validation samples. This means that there is no need to access the OOD data. The authors show that  the CDR framework is compatible with different density estimators (CDR_KDE, CDR, Maha).
The performed experiments on benchmark datasets (CIFAR-10, CIFAR-100, ImageNet) show the effectiveness in both supervised and self-supervised learning scenarios and competitiv/superior results compared to existing methods including the MSP, Energy score, and GEM.
The paper also provides theoretical foundations for the proposed approach.

**Strengths:**

First the paper addresses a practical, real-world scenario where there is only access to pre-trained model weights and limited validation data.  The theoretical foundation on presenting a mathematically grounded approach to OOD detection with clear justification for its effectiveness. The proposed CDR framework demonstrates versatility, being compatible with different density estimators like KDE and Mahalanobis, which allows for adaptation to various situations and requirements.
Another strong point is the introduction of an innovative temperature scaling technique that works without requiring OOD samples, showing a good performance across a wide range of datasets and architectures.
The paper shows effective performance in both supervised and self-supervised learning scenarios.

**Weaknesses:**

The most significant theoretical limitation lies in its core assumption that the marginal distribution $p(z)$ is a good approximation of the true data distribution $p(x)$. The provided theoretical justification for why this approximation should hold and under what conditions it might break down is insufficient.
The evaluations are reach in computer vision while other application domains are neglected. This focus on computer vision raises questions about the generalizability of the method to other types of data (problem spaces). Also, the authors need to discuss the computational overhead of their approach compared to baseline methods, particularly for the CDR_KDE variant which likely has higher computational requirements due to its non-parametric nature.
There is the need for more  exploration of why CDR performs poorly in scenarios where Mahalanobis distance underperforms.
The relationship between temperature scaling and model calibration needs more deeper theoretical analysis to explain why it works so well in practice.
The authors use  averaging CDR scores across classes. It shows good results in practical examples, however, there is need for sufficient theoretical motivation.
The practical implementation considerations, including memory requirements are not thoroughly discussed.

**Questions:**

1- Could authors provide a more rigorous justification for why $p(z)$ serves as a good approximation of $p(x)$? (This is a fundamental assumption of the proposed method)

2- What are the concrete computational trade-offs (memory, scaling behavior) between CDR_KDE and CDR_Maha compared to baseline methods?

3- As indicated the CDR performs poorly when Mahalanobis distance underperforms. Could the authors provide a more detailed analysis of when and why this occurs?

---

> ### Author Response · Authors · 2024-11-20
>
> We would like to thank the reviewers for taking the time to read our paper and provide valuable feedback. We are encouraged by the comments recognizing that the proposed CDR method is novel, theoretically driven, and versatile (reviewers 9Qn6, 5AjN, and hsYo); that the temperature scaling method is effective without OOD samples (reviewers 9Qn6 and hsYo); and that the proposed method performs well empirically (reviewers 9Qn6, hsYo, GZ1a, and 5AjN). Below, we address the comments, suggestions, and questions in detail.
>
> ### **Why is $p(\boldsymbol{z})$ a good approximation of $p(\boldsymbol{x})$?**
> This is an excellent question. Unfortunately, there is currently no definitive answer to this and remains as an open problem to the best of our knowledge. This assumption is commonly made in the post-hoc density estimation literature for OOD detection, as seen in prior methods such as KNN [1], the energy score [2], and GEM [3]. The intuition is that, for a classifier to perform accurately, $p(\boldsymbol{z})$ should retain the most critical information about $p(\boldsymbol{x})$. However, this assumption may be inaccurate in practice, potentially leading to suboptimal performance.
>
> Beyond the post-hoc OOD detection setup, we believe the CDR score can serve as a general framework for density estimators. For example, instead of relying on a trained classifier to obtain $p(\boldsymbol{z}|y)$, an independently trained likelihood-based neural network could directly estimate $p(\boldsymbol{x}|y)$. In this case, we hypothesize that the CDR framework could effectively combine $p(\boldsymbol{x}|y)$ and $p(y|\boldsymbol{x})$ to estimate $p(\boldsymbol{x})$ more accurately. We leave this for future exploration.
>
> ### **Additional Experiments with Other Application Domains**
> We thank the reviewer for raising this concern. OOD detection benchmarks and tasks have predominantly focused on computer vision. To further validate the applicability of our method, we included additional experiments with NLP datasets. Details are available in Appendix C (we have added pointers in the main paper to guide readers to this section). For simplicity, the features for these experiments were directly obtained from ADBench [4] without modifications. To our knowledge, this is the first use of these NLP datasets for OOD detection evaluation. As seen from the results, our conclusions remain consistent across domains. We emphasize that we used an identical setup for all experiments to ensure fairness and reproducibility.
>
> ### **Concrete Computational Trade-offs Between KDE and Mahalanobis Estimators**
> We acknowledge that the KDE estimator is computationally more expensive than the Mahalanobis estimator during inference. Specifically:
>
> 1. During training, the Mahalanobis estimator requires an additional step to compute the mean and covariance matrix from the validation dataset.
> 2. During inference, the computational complexity of KDE scales linearly with the number of samples, while the Mahalanobis estimator has constant complexity with respect to the sample size.
>
> However, the number of samples used is kept very small in our experiments. For example, in the ImageNet experiments, we use only 25 samples per class to compute both the mean and variance for the Mahalanobis estimator and the KDE score. Additionally, since the KDE score is computed in $\boldsymbol{z}$, we precompute the latent representation vectors $\boldsymbol{z}$ for all validation samples. This allows for efficient computation of the CDR score, as it only involves a single forward pass through the classifier and some matrix-vector multiplications.
>
> We have included a discussion on these computational trade-offs in the paper.
>
> ### **Why Does CDR Perform Poorly When the Mahalanobis Estimator Fails?**
> The proposed CDR score can be understood through its decomposition in Equation 2 of the paper. In this decomposition, the CDR score acts as a correction to the energy score using an "average generative conditional ratio" (GCR) term. Since the GCR term depends on the Mahalanobis estimator, poor performance of the Mahalanobis estimator directly impacts the correction, potentially harming OOD detection performance.

---

> ### Author Response · Authors · 2024-11-20
>
> ### **Relationship Between Temperature Scaling and Model Calibration**
> Tuning temperatures is straightforward with some labeled OOD samples. However, in this work, we consider a more challenging yet realistic scenario where no OOD samples are available for hyperparameter tuning.
>
> The guiding principles of our temperature scaling method are twofold:
> 1. Calibration of $p(y|\boldsymbol{z})$: Neural networks are often overconfident, and tuning the temperature can improve log-likelihood [5], which we hypothesize enhances OOD detection performance. This is confirmed by our experiments in Appendix E.
> 2. Balancing $p(y|\boldsymbol{z})$ and $p(\boldsymbol{z}|y)$: For CDR to be most effective, the scales of $p(y|\boldsymbol{z})$ and $p(\boldsymbol{z}|y)$ should not differ significantly. Empirically, these scores are often orders of magnitude apart, and summing them naively provides little improvement. To address this, we tune the temperature of $p(\boldsymbol{z}|y)$ while fixing $p(y|\boldsymbol{z})$, and add a regularization term to minimize the $L_1$ difference between their scales.
>
> ### **Motivation for Averaging CDR Scores**
> Under perfect estimation of $p(\boldsymbol{z}|y)$ and $p(y|\boldsymbol{z})$, all individual CDR scores across classes are identical. In practice, however, taking the average helps reduce estimation variance, leading to a better approximation of $p(\boldsymbol{z})$ and improved OOD detection performance.
>
> ### **Additional Post-Hoc OOD Detection Methods**
> Lastly, we have implemented two recently proposed post-hoc OOD detection methods for comparison. Preliminary results on CIFAR datasets are provided in the table below. In general, the proposed CDR score performs competitively against these methods. We are still finalizing the experiments and will update the paper with the complete results once they are available.
>
> | Inlier Data   | CIFAR10                    |                           |                           |                           | CIFAR100                  |                           |                           |                           |
> |---------------|----------------------------|---------------------------|---------------------------|---------------------------|---------------------------|---------------------------|---------------------------|---------------------------|
> | OOD Data      | Average OOD               |                           | Average Hard-OOD          |                           | Average OOD               |                           | Average Hard-OOD          |                           |
> | Metrics       | FPR95                     | AUROC                     | FPR95                     | AUROC                     | FPR95                     | AUROC                     | FPR95                     | AUROC                     |
> | GEN (Liu et al, CVPR 2023) | 37.8         | 92.45                     | 47.98                     | 89.39                     | 81.7                      | 77.60                     | 83.84                     | 74.42                     |
> | fDBD (Liu and Qin, ICML 2024) | 28.15     | 95.23                     | 45.50                     | 90.16                     | 59.05                     | 87.11                     | 77.38                     | 77.26                     |
> | CDR_KDE       | 14.68                     | 97.01                     | **38.65**                 | **91.05**                 | 30.52                     | 92.69                     | **69.87**                 | **77.91**                 |
> | CDR_Maha      | **10.53**                 | **97.72**                 | 42.36                     | 90.21                     | **19.55**                 | **96.01**                 | 71.33                     | 75.90                     |
>
> Please do not hesitate to provide further comments or suggestions. Thank you once again for your valuable feedback!
>
> [1] Sun, Yiyou, et al. "Out-of-distribution detection with deep nearest neighbors." International Conference on Machine Learning. PMLR, 2022.
>
> [2] Liu, Weitang, et al. "Energy-based out-of-distribution detection." Advances in neural information processing systems 33 (2020): 21464-21475.
>
> [3] Morteza, Peyman, and Yixuan Li. "Provable guarantees for understanding out-of-distribution detection." Proceedings of the AAAI Conference on Artificial Intelligence. Vol. 36. No. 7. 2022.
>
> [4] Han, Songqiao, et al. "Adbench: Anomaly detection benchmark." Advances in Neural Information Processing Systems 35 (2022): 32142-32159
>
> [5] Guo, Chuan, et al. "On calibration of modern neural networks." International conference on machine learning. PMLR, 2017.
>
> [6] Liu, Litian, and Yao Qin. "Fast Decision Boundary based Out-of-Distribution Detector." Forty-first International Conference on Machine Learning, 2024.
>
> [7] Liu, Xixi, Yaroslava Lochman, and Christopher Zach. "Gen: Pushing the limits of softmax-based out-of-distribution detection." CVPR 2023.

---

### Official Review · Reviewer_9Qn6 · 2024-11-02

**Soundness:** 3
**Presentation:** 3
**Contribution:** 3
**Rating:** 6
**Confidence:** 2

**Summary:**

This paper introduces the Conditional Density Ratio (CDR) score, a post hoc method for deep outlier detection, which combines a class-conditional generative model with a discriminative classifier to estimate densities without marginalizing over classes. CDR enables effective detection of out-of-distribution (OOD) samples by leveraging both models and incorporating a novel temperature scaling technique that does not require OOD samples for calibration. The proposed method works with commonly used density estimators like the Mahalanobis distance and kernel density estimation, achieving competitive performance across a wide range of OOD benchmarks.

**Strengths:**

s1. The paper introduces a novel and practical approach to model both aleatoric and 1. The studied problem is important and practice, and the paper is well-organized with well presentation.

2. The proposed Conditional Density Ratio (CDR) score can effectively integrate generative and discriminative models, providing a principled approach to outlier detection without the need for marginalization over classes.

3. The paper introduces a novel temperature scaling technique that allows for automatic calibration of model parameters without requiring OOD samples.  And experiments on the high-dimensional datasets demonstrate the effectiveness of the proposed score.

**Weaknesses:**

w1. The proposed model is mainly tested in high-dimensional data, it would be better to test on more datasets as in [1], to see whether it can perform well in small datasets:

[1] ADBench: Anomaly Detection Benchmark, https://arxiv.org/abs/2206.09426


w2. Since the function g(x) depends on \lambda, in practice, how do we decide lambda for each dataset for better OOD detection?

**Questions:**

Please see the weaknesses above.

---

> ### Author Response · Authors · 2024-11-20
>
> We would like to thank the reviewers for taking the time to read our paper and provide valuable feedback. We are encouraged by the comments recognizing that the proposed CDR method is novel, theoretically driven, and versatile (reviewers 9Qn6, 5AjN, and hsYo); that the temperature scaling method is effective without OOD samples (reviewers 9Qn6 and hsYo); and that the proposed method performs well empirically (reviewers 9Qn6, hsYo, GZ1a, and 5AjN). Below, we address the comments, suggestions, and questions in detail.
>
> ### **Additional Experiments with ADBench**
> We thank the reviewer for suggesting additional benchmarking experiments using the ADBench dataset. We invite the reviewer to refer to the additional experiments conducted with the NLP datasets, as described in Appendix C (we have added comments in the main paper to guide readers to this section). The features for these experiments were directly obtained from the ADBench datasets without modification. As shown in the results, similar conclusions can be drawn.
>
> While we appreciate the suggestion to incorporate further experiments using ADBench datasets to demonstrate the effectiveness of the proposed method, many datasets in ADBench are not well suited to our problem setup, which involves training a classifier to obtain both $p(y|\boldsymbol{z})$ and $p(\boldsymbol{z}|y)$. Most ADBench datasets are designed for anomaly detection and are inherently unlabeled, containing only inlier and outlier samples.
>
> ### **How to Choose Lambda for the OOD Function**
> Lambda ($\lambda$) can be selected either by using a small labeled OOD dataset or by setting a threshold based on the inlier sample distribution. For instance, the 0.5% tail of the inlier sample distribution can be used as the cutoff threshold.
>
> We acknowledge that choosing $\lambda$ for OOD detection can be challenging in practice. However, we would like to emphasize that the evaluation metrics we use (FPR95 and AUROC) are independent of the choice of $\lambda$.
>
> ### **Additional Post-Hoc OOD Detection Methods**
>
> Lastly, we have implemented two recently proposed post-hoc OOD detection methods for comparison. Preliminary results on CIFAR datasets are provided in the table below. In general, the proposed CDR score performs competitively against these methods. We are still finalizing the experiments and will update the paper with the complete results once they are available.
>
> | Inlier Data   | CIFAR10                    |                           |                           |                           | CIFAR100                  |                           |                           |                           |
> |---------------|----------------------------|---------------------------|---------------------------|---------------------------|---------------------------|---------------------------|---------------------------|---------------------------|
> | OOD Data      | Average OOD               |                           | Average Hard-OOD          |                           | Average OOD               |                           | Average Hard-OOD          |                           |
> | Metrics       | FPR95                     | AUROC                     | FPR95                     | AUROC                     | FPR95                     | AUROC                     | FPR95                     | AUROC                     |
> | GEN (Liu et al, CVPR 2023) | 37.8         | 92.45                     | 47.98                     | 89.39                     | 81.7                      | 77.60                     | 83.84                     | 74.42                     |
> | fDBD (Liu and Qin, ICML 2024) | 28.15     | 95.23                     | 45.50                     | 90.16                     | 59.05                     | 87.11                     | 77.38                     | 77.26                     |
> | CDR_KDE       | 14.68                     | 97.01                     | **38.65**                 | **91.05**                 | 30.52                     | 92.69                     | **69.87**                 | **77.91**                 |
> | CDR_Maha      | **10.53**                 | **97.72**                 | 42.36                     | 90.21                     | **19.55**                 | **96.01**                 | 71.33                     | 75.90                     |
>
> Please do not hesitate to provide further comments or suggestions. Thank you once again for your valuable feedback and support!
>
> [1] Liu, Litian, and Yao Qin. "Fast Decision Boundary based Out-of-Distribution Detector." Forty-first International Conference on Machine Learning, 2024.
>
> [2] Liu, Xixi, Yaroslava Lochman, and Christopher Zach. "Gen: Pushing the limits of softmax-based out-of-distribution detection." Proceedings of the IEEE/CVF Conference on Computer Vision and Pattern Recognition. 2023.

---

> > ### Comment · Reviewer_9Qn6 · 2024-11-24
> >
> > Thanks for the response and additional experiments.

---

> > > ### Author Response · Authors · 2024-11-25
> > >
> > > Thank you so much for reading our response! Please do not hesitate if you have any further concerns!

---

### Meta-Review · Area_Chair_zZxu · 2024-12-19

**Metareview:**

The authors propose a novel method for Out-of-Distribution (OOD) estimation in scenarios where training examples are unavailable, and only a small validation set of in-distribution data is provided. Their approach combines density estimation of the latent representation with a conditional classifier to determine whether a sample is OOD.

There was significant disagreement among the reviewers. The more critical reviewers pointed out a lack of novelty and that the method was only compared to a subset of algorithms for OOD detection. The authors acknowledged this critique, clarifying that their method does not claim state-of-the-art (SOTA) performance within the broader OOD detection framework, but rather in the specific scenario they address. Meanwhile, the more favorable reviewers were inclined to support the paper's presentation at ICLR.

This paper can be considered borderline. While it provides a solution for a specific OOD detection scenario, further comparisons with SOTA methods—even under different assumptions—would strengthen its case. Additionally, evaluating the breadth and impact of the proposed method's assumptions would enhance the paper's overall contribution.

**Additional Comments On Reviewer Discussion:**

The authors engaged with the reviewers during the discussion phase but were unable to convince one of them. While the authors attempted to argue that Reviewer GZ1a's comments were unreasonable, I found those comments to be fair and valid. Additionally, the two reviewers who gave a score of 6 were not particularly enthusiastic either.

The paper does have some merit and could be accepted, but its rejection would not represent a significant loss to the field.

---

### Decision · Program_Chairs · 2025-01-22

Reject